 

# Patterned perturbation of inhibition can reveal the dynamical structure of neural processing

**Sadra Sadeh\*, Claudia Clopath**

Bioengineering Department, Imperial College London, London, United Kingdom

**Abstract** Perturbation of neuronal activity is key to understanding the brain's functional properties, however, intervention studies typically perturb neurons in a nonspecific manner. Recent optogenetics techniques have enabled patterned perturbations, in which specific patterns of activity can be invoked in identified target neurons to reveal more specific cortical function. Here, we argue that patterned perturbation of neurons is in fact necessary to reveal the specific dynamics of inhibitory stabilization, emerging in cortical networks with strong excitatory and inhibitory functional subnetworks, as recently reported in mouse visual cortex. We propose a specific perturbative signature of these networks and investigate how this can be measured under different experimental conditions. Functionally, rapid spontaneous transitions between selective ensembles of neurons emerge in such networks, consistent with experimental results. Our study outlines the dynamical and functional properties of feature-specific inhibitory-stabilized networks, and suggests experimental protocols that can be used to detect them in the intact cortex.

## Introduction

Our understanding of sensory processing in the cortex is moving beyond simple stimulus-response analysis of single cells, and the role of networks of neurons is becoming increasingly more important in this process. Such network effects are, however, difficult to study by recording single cells in isolation or even by measuring the activity of populations of neurons, if our interaction with the circuitry is only observational. A more promising approach is to record and perturb the activity of neuronal populations at the same time. However, theory and computational modelling are needed to guide such endeavours, due to the complexity of the high-dimensional space in which experiments can be performed.

Experimentally, optical stimulation of neurons has emerged in recent years as a powerful technique to perturb neuronal activity in order to study brain function and dysfunction (**Boyden et al., 2005**; **Fenno et al., 2011**; **Yizhar et al., 2011**; **Boyden, 2015**). Optogenetic perturbations have provided us with an effective means to interrogate neuronal connectivity and dynamics in action. Despite these advances, typical perturbation paradigms are still mainly limited to simple profiles of activation or deactivation, hindering a more precise stimulation of neuronal activity which would be necessary to study specific brain functions. Although increasing number of perturbation studies are targeting specific subgroups of excitatory or inhibitory neurons with genetic markers, within each subpopulation still nonspecific perturbation patterns (either in terms of their spatiotemporal profile or functional features) are primarily used to interrogate the circuitry (**Atallah et al., 2012**; **Lien and Scanziani, 2013**; **Pfeffer et al., 2013**; **Reinhold et al., 2015**; **Kato et al., 2017**). However, recent advances in optogenetics are making a closed-loop, all-optical interrogation of neuronal circuitry, where neuronal activity can be measured and perturbed simultaneously by optical tools, more achievable (**Emiliani et al., 2015**; **Ronzitti et al., 2017**; **Zhang et al., 2018**), hence paving the way for more specific, patterned perturbations. Several recent studies have in fact used such patterned

**\*For correspondence:**
s.sadeh@imperial.ac.uk

**Competing interests:** The authors declare that no competing interests exist.

perturbations to study cortical function with more precision, for example by recording neurons involved in specific tasks (e.g. natural visual processing or memory tasks) and reactivating them in order to instigate or suppress the relevant behaviour (*Chernov et al., 2018*; *Carrillo-Reid et al., 2019*; *Marshel et al., 2019*; *Russell, 2019*; *Tran et al., 2019*).

Optogenetic techniques have particularly been exploited to cast light on the regime of operation of cortical networks, especially regarding the role of feedforward versus recurrent pathways. A purely feedforward regime of information processing relies on the transmission of information between layers, while a recurrent regime adds an extra level of processing within each layer. The actual architecture of the neocortex seems to favour the latter, as the recurrent connections outnumber the feedforward inputs (*Peters and Payne, 1993*). Strong recurrent connections between principal cells can enable them to perform many useful computations (such as signal amplification, denoising and pattern completion), but this can also lead to runaway excitation and hence unstable dynamics in the brain. It has therefore been suggested that neocortical networks operate in an inhibition-stabilized regime (inhibitory-stabilized networks, or ISNs), where the high gain of cortical recurrent excitation is stabilized, or balanced, by a potent recurrent inhibition (*van Vreeswijk and Sompolinsky, 1996*; *van Vreeswijk and Sompolinsky, 1998*; *Tsodyks et al., 1997*; *Ozeki et al., 2009*). In other words, under ISN regimes of activity, cortical networks would not be stable without inhibition and therefore undergo runaway activity.

The long-lasting question about whether cortical networks operate as inhibitory-stabilized networks has been reconsidered in the wake of new optogenetic techniques. It has been probed by measuring the presence or absence of paradoxical responses of inhibitory neurons upon perturbation (*Tsodyks et al., 1997*; *Ozeki et al., 2009*). If the network is primarily feedforward driven, increasing the input to inhibitory neurons should lead to an intuitive increase in their responses. On the other hand, in ISNs, dominated by strong recurrent interaction of excitation and inhibition, increasing the input to inhibitory neurons *decreases* their activity, in a non-intuitive direction (hence 'the paradoxical effect'). Intuitively, this is because, due to the strong recurrence, increasing the input to inhibitory neurons increases inhibition to excitatory neurons, which in turn leads to an overall decrease in the drive to inhibitory neurons. Although earlier studies failed to report the presence of such paradoxical effects in mouse visual cortex (*Atallah et al., 2012*), more recent optogenetics experiments were able to observe them in the visual cortex of awake mice (*Adesnik et al., 2012*) as well as in auditory, somatosensory, and motor cortices (*Kato et al., 2017*; *Sanzeni, 2019*); but see *Mahrach et al., 2020*).

The conflicting results in different experiments could partly be related to simplifications in the studies of ISNs. Like optogenetic manipulations, ISN properties too have mainly been studied for simplified scenarios, both experimentally and theoretically. For example, the original theoretical model of ISNs reduced the populations of excitatory and inhibitory neurons to single nodes in the analysis (*Tsodyks et al., 1997*). New modelling studies have started to account for more biological realism, for example by studying the effect of including multiple subtypes of inhibitory neurons (*Litwin-Kumar et al., 2016*) or nonlinear neuronal transfer functions (*Ahmadian et al., 2013*; *Rubin et al., 2015*; *Hennequin et al., 2018*). Another extension of the original model to include more biological details (*Sadeh et al., 2017*; *Gleeson et al., 2019*) concluded that observation of the paradoxical effect may depend on the size of perturbation of the inhibitory population; a result which was in fact corroborated by subsequent experimental studies (*Li et al., 2019*; *Sanzeni, 2019*). Further theoretical and experimental exploration of ISNs can, therefore, reveal new insights about the dynamical and functional properties of cortical networks.

An important aspect that has remained understudied in the study of ISNs is the functional specificity of connections, as reported in cortical networks. For instance, a highly specific connectivity between excitatory neurons has recently been reported in mouse visual cortex, whereby neurons with similar functional properties (e.g., in terms of their receptive fields) are strongly connected (*Ko et al., 2011*; *Ko et al., 2013*; *Cossell et al., 2015*; *Lee et al., 2016*). Simulation of networks with experimentally recorded weights of specific connections within the excitatory subnetwork, however, suggests that large-scale networks with such connectivity profiles would be unstable (*Znamenskiy et al., 2018*). This raises the question of which mechanisms are recruited by the cortex to stabilize the unstable dynamics of functional subnetworks. One possibility is that, similar to the balancing of the nonspecific dynamics by recurrent inhibition, the specific dimension of the excitatory connectivity is also balanced by a matching specific inhibition, therefore forming subnetworks

of excitatory and inhibitory neurons, all coding for similar visual features. A recent study has in fact found evidence for the presence of such functionally specific excitatory-inhibitory (E→I and I→E) connectivity in visual networks of rodents (*Znamenskiy et al., 2018*), suggesting that E-I subnetworks may also exist in addition to excitatory subnetworks.

The possibility of specific inhibitory feedback balancing highly specific and potentially unstable excitatory subnetworks adds another dimension to inhibitory stabilization in cortical networks. We therefore aimed to study such 'functionally specific ISNs' (or specific ISNs in short) here, and to characterize the dynamical signatures and functional properties arising in these networks. We found that, across model networks at multiple scales and with different degrees of biological realism, specific ISNs can be identified by a 'specific paradoxical effect'. That is, by exciting only the inhibitory neurons within a subnetwork where all neurons respond to similar stimuli, the activity of the inhibitory neurons in the subnetwork decreases. Moreover, the inhibitory neurons that receive more perturbation (more excitation) decrease their activity more. This dynamical fingerprint would only be observed if the network is probed with patterned perturbations aligned with functional properties of neurons (i.e., perturbations are delivered within a subnetwork rather than distributed across subnetworks). Moreover, this dynamical regime was concomitant with the emergence of spontaneous transitions between selective neuronal activity patterns, a feature that has been reported experimentally in different species (*Kenet et al., 2003*; *Miller et al., 2014*). Our study thus highlights the importance of patterned perturbation in probing the operating regime of cortex and its dynamical and functional properties.

## Results

### Optogenetic perturbation to detect inhibitory stabilized networks

Networks with high excitatory connectivity which are balanced by a strong inhibitory feedback are called inhibitory stabilized networks (ISNs). The strong recurrent interaction between excitation and inhibition in these networks manifests itself in a dynamical signature, which is the paradoxical change in the activity of inhibitory neurons upon their perturbation (*Tsodyks et al., 1997*). That is, if the input to inhibitory neurons are increased, the activity of inhibitory neurons is *decreased*, in a paradoxical manner. ISNs –and their associated paradoxical effects– have mainly been studied for simple connectivity profiles, by assuming a uniform connectivity between excitatory and inhibitory subpopulations. Recent experimental studies, on the other hand, have revealed a highly specific connectivity profile between excitatory neurons (*Ko et al., 2011*; *Cossell et al., 2015*), which is likely to be balanced by a specific inhibitory feedback (*Znamenskiy et al., 2018*). However, the precise effect of such specific connectivity (*Figure 1A*) on dynamic responses of ISNs remains understudied, both theoretically and experimentally.

Experimental optogenetics paradigms typically perturb inhibitory neurons randomly and uniformly to probe the presence of ISN networks ('non-specific' perturbation; *Figure 1—figure supplement 1*). This is performed by, for example, reducing the input to many inhibitory neurons expressing channelrhodopsins and measuring the change in the activity of the perturbed inhibitory population (*Atallah et al., 2012*) or the change in the input to pyramidal cells (*Adesnik, 2017*). If the average change is in the opposite direction of perturbation, that is an overall increase in the activity when reducing the input (*Adesnik, 2017*; *Kato et al., 2017*), the effect of perturbation is called paradoxical, which is the signature of ISN regimes of activity (*Tsodyks et al., 1997*; *Figure 1B,C*; *Figure 1—figure supplement 1A,B*).

An alternative way to experimentally perturb inhibitory neurons is to increase or decrease their activity according to a specific pattern (specific or 'patterned' perturbation) (*Figure 1B*). Consider a simplified scenario where neurons are grouped into subnetworks responding preferentially to a one-dimensional feature like orientation of edges. Patterned perturbations are now delivered such that neurons within a specific subnetwork (e.g. the subnetwork best responding to 90°) are perturbed the most, while neurons belonging to other subnetworks are perturbed based on their functional distance in the feature space (i.e. the subnetwork best responding to 0° receives the least perturbation, as it has the highest distance in preferred orientation) (*Figure 1B*).

If inhibitory stabilization is specific to subnetworks, that is the excessive excitatory activity within each subnetwork is balanced by a matching specific inhibition, specific ISNs will emerge (*Figure 1—*

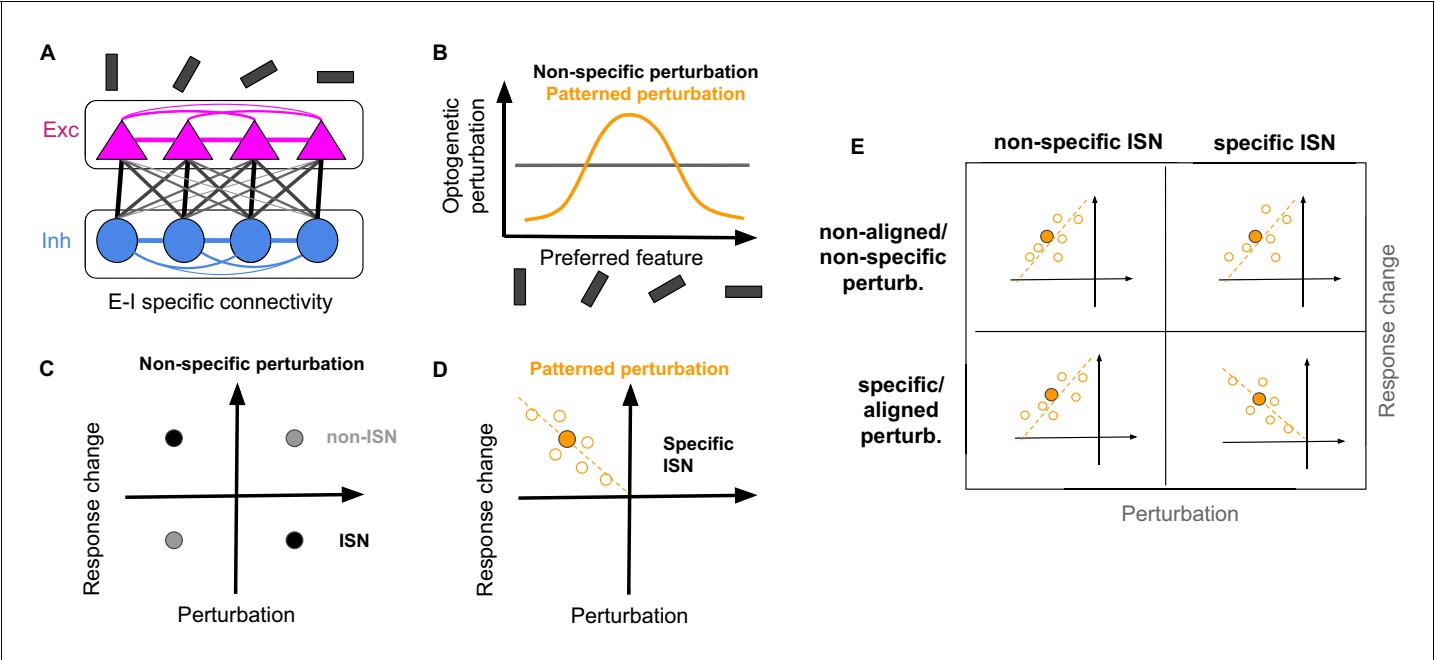

**Figure 1.** Random and patterned perturbation of inhibition to asses inhibitory stabilized networks. (A) Schematic of network with specific EI connectivity. (B) Optogenetic perturbation of inhibition can either be non-specific (independent of preferred features of the perturbed inhibitory neurons) or patterned (specifically addressing inhibitory neurons with similar feature selectivity). (C) Non-specific perturbation only reveals the average paradoxical effect in an ISN (black circle), by revealing that the average activity of perturbed inhibitory neurons is changing in the opposite direction of perturbations. In non-ISNs (gray circle), perturbations and response changes have the same sign. (D) Patterned perturbation of specific ISNs reveals specific paradoxical effects, where inhibitory neurons with more (negative) perturbations show more (positive) response changes. The negative correlation of perturbations and response changes is indicated by the dashed line with a negative slope. The mean activity of inhibitory neurons (filled orange circle) still shows the non-specific paradoxical effect, similar to (C). The specific paradoxical effect is, however, reflected in the slope of the pattern, and cannot be distinguished by the mean activity. (E) Expected patterns for the response changes of inhibitory neurons versus their perturbations under four scenarios: non-specific or specific perturbation of non-specific of specific ISNs. Similar to (D), open circles denote the response change of individual inhibitory neurons upon perturbation, the filled circle is the average across those neurons, and the dashed line indicates the slope of correlation. While all combinations show non-specific paradoxical effect (reflected in the similar behaviour of the mean response), only specific perturbation of specific ISNs reveal the specific paradoxical effect (reflected in the negative slope of the line).

The online version of this article includes the following figure supplement(s) for figure 1:

**Figure supplement 1.** Paradoxical effects of perturbing inhibition in ISNs and specific ISNs.

figure supplement 1C). In contrast to uniform (or non-specific) ISNs, non-specific perturbation is not enough to detect specific ISNs; instead, patterned perturbation (as described above) is needed. The specific ISN is observed, if there is an anticorrelation between the perturbation of inhibitory neurons and their response change (*Figure 1D*). We refer to this as the 'specific paradoxical effect', which distinguishes specific ISNs from both non-ISNs and non-specific ISNs (*Figure 1—figure supplement 1*). Note that this signature would not be revealed by the average activity of the perturbed inhibitory population, as in both non-specific and specific ISNs, the mean activity of perturbed inhibitory neurons shows the paradoxical effect (c.f. *Figure 1C,D*). The specific paradoxical effect, in contrast, manifests itself in the second-order relation of the input perturbation and output activity (i.e. the negative slope), and only if the pattern of perturbation is aligned with functionally specific connectivity of the network (*Figure 1—figure supplement 1C*).

Four possible scenarios are thus conceivable out of the interaction of nonspecific/specific perturbations and nonspecific/specific ISNs (*Figure 1E*). Only in one of them, namely specific perturbations of specific ISNs, the negative slope would be expected. In all other combinations, the extra perturbations around the mean change the response of inhibitory neurons in the same direction. That is, if an inhibitory neuron is perturbed more than the mean perturbation, it would obtain an extra response change more than the mean value; and vice versa. This is because the vector of residual perturbations around the mean is not aligned with the feature-specific dynamics of the network (in

non-specific perturbations), or simply because the network lacks such feature-specific dynamics (in non-specific ISNs). The result is a *positive* slope: although the mean response changes still show the non-specific paradoxical effect, the differential response changes around the mean behave in the same direction as differential perturbations.

## Specific perturbation to detect feature-specific inhibitory stabilization

To study how specific ISNs behave under different patterns of perturbations, we first simulated large-scale rate-based networks of excitatory and inhibitory neurons (*Figure 2*). Neurons were assigned initial preferred orientations and the weight matrix was constructed such that pairs of neurons (E-to-E, E-to-I, I-to-E and I-to-I) with similar preferred orientations were more strongly connected to each other (see Materials and methods). Under both paradigms of perturbation (perturbations independent of neuron's preferred orientation or in accordance to it; *Figure 1B*), the average activity of the network showed the conventional paradoxical effect. That is, negative perturbations of inhibitory neurons (induced by *decreasing* the input to them) *increased* the activity of both inhibitory and excitatory populations, on average (*Figure 2A*).

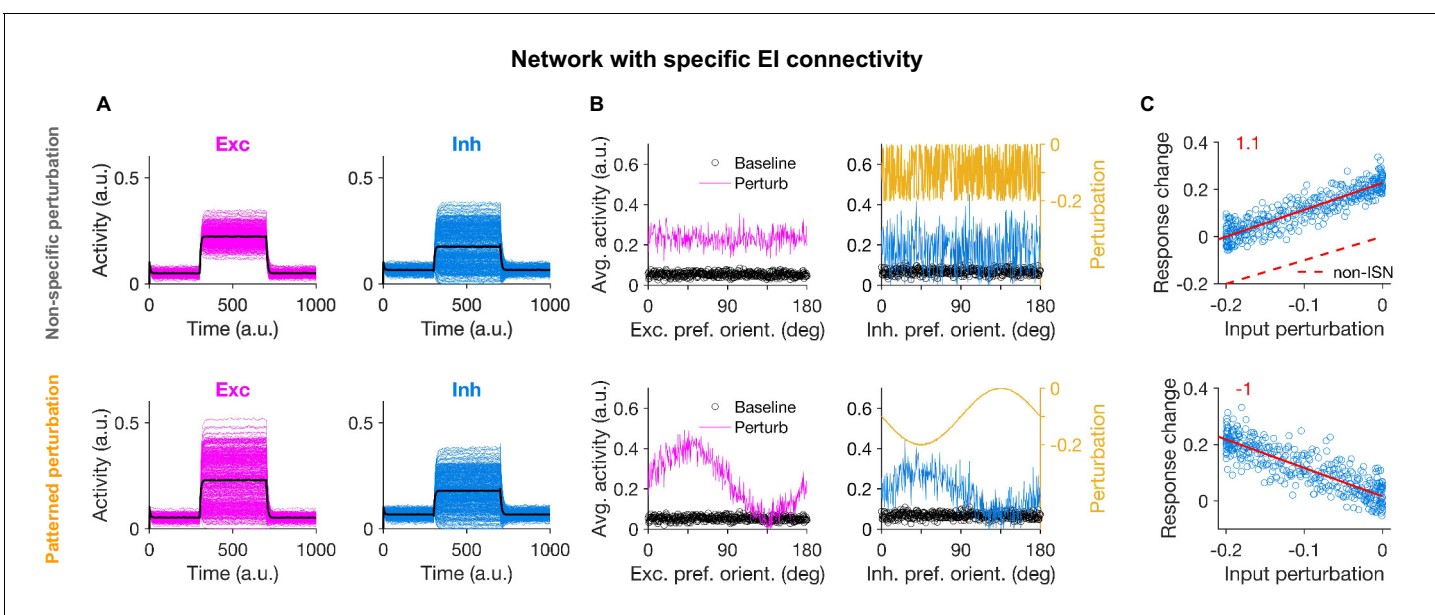

**Figure 2.** Patterned perturbation of inhibition is necessary to reveal specific inhibitory stabilization. (**A**) Activity of individual excitatory (magenta) and inhibitory (blue) neurons, and their average activity (black), in the baseline state and during perturbation (from T = 300 to 700) applied by reducing the input to inhibitory neurons (see Materials and methods). Perturbations are induced either non-specifically (upper) or in a patterned manner (lower), where the decrease in the input is proportional to the preferred orientation of respective inhibitory neurons (*Figure 1B*). Both modes of optogenetic perturbations lead to the non-specific paradoxical effect, as reducing the input to inhibitory neurons paradoxically increases their average activity. (**B**) Average activity of individual excitatory and inhibitory neurons during baseline (black dots) and perturbed states as a function of their preferred orientations. The amount of perturbation of the input to each inhibitory neuron is plotted on the right side (orange) for comparison. The non-specific perturbation is the same pattern as the specific perturbation but randomized over neurons. (**C**) Response change as a result of perturbation (calculated as the difference in the average activity of neurons in the perturbed states compared to the baseline) for inhibitory neurons as a function of their input perturbation. This is shown for specific EI networks in response to nonspecific perturbation (upper) and patterned perturbations (lower). Solid red lines show the best fitted regression line to the data points, with the slope of the line indicted on top (red). The dashed line in (upper) illustrates how a non-ISN would have behaved under a similar perturbation (*Figure 2—figure supplement 2*).

The online version of this article includes the following figure supplement(s) for figure 2:

**Figure supplement 1.** Lack of specific paradoxical effect in networks without specific EI connectivity.
**Figure supplement 2.** Non-paradoxical effect in non-ISN networks.
**Figure supplement 3.** Specific paradoxical effect with positive perturbations.
**Figure supplement 4.** Patterned perturbation of inhibitory neurons in networks with fewer inhibitory neurons and broader inhibitory connectivity.
**Figure supplement 5.** Patterned perturbation of inhibitory neurons in networks with heterogeneous specificity of excitatory and inhibitory connections.
**Figure supplement 6.** Feature-specific paradoxical effects in networks with nonlinear transfer functions.
**Figure supplement 7.** Feature-specific paradoxical effects in networks with partial perturbation of inhibition.

The difference between the two types of perturbation, however, became evident when the response changes were plotted in terms of the feature-selectivity of neurons (*Figure 2B*). Nonspecific perturbations increased the activity of inhibitory neurons without any obvious pattern in terms of their preferred orientations (*Figure 2B*). In contrast, patterned perturbations invoked inverse patterns of response changes, such that inhibitory neurons with larger (more negative) perturbations showed higher *increase* in their activity (*Figure 2B*, lower). This 'feature-specific paradoxical effect' was clearly evident in the negative correlation between the response change of inhibitory neurons and their input perturbations (*Figure 2C*, lower), and could be quantified by the negative slope of the regression line fitted to the data points (*Figure 2C*). We used this metric in the rest of the study to quantify the presence of feature-specific ISNs.

The negative slope was missing for the nonspecific pattern of perturbation (*Figure 2C*, upper), which was a pattern generated by randomizing the specific pattern (*Figure 2B*, upper). This indicates that the precise feature-specific pattern of perturbation is crucial to reveal the specific paradoxical effect. The specific ISN behaviour was also not observed in networks without feature-specific EI connectivity during either non-specific or patterned perturbations, despite the presence of the non-specific paradoxical effect (*Figure 2—figure supplement 1*). As expected (*Figure 1E*), the slope was positive, since in the absence of specific paradoxical effects, inhibitory neurons show higher relative responses for the differential perturbations around the mean level. Since the mean level is nevertheless showing the nonspecific paradoxical effect, this positive slope leads to a counterintuitive observation that inhibitory neurons with no perturbations show some positive response changes (*Figure 2C*, upper).

Note that, while the nonspecific paradoxical effect is reflected in the mean population behaviour of the network (i.e., in the average response change and the average perturbation of inhibitory population), the specific paradoxical effect can be inferred from the second-order statistics. That is, the slope of the relationship between the input perturbations and output response changes across inhibitory neurons should be considered, rather than their population average values. In all three conditions discussed above (*Figure 2C*, upper and *Figure 2—figure supplement 1C*), the mean activity of the inhibitory population shows the conventional, nonspecific paradoxical effect. For comparison, such a nonspecific paradoxical effect would be absent in non-ISN networks, as their inhibition shows average response changes in the same direction as input perturbations (*Figure 2—figure supplement 2*, and dashed line in *Figure 2C*, upper). Lack of an inverse second-order relationship (negative slope) is, however, present in all these four conditions (specific ISNs with randomized patterns (*Figure 2C*), nonspecific ISNs in response to patterned or randomized perturbations (*Figure 2—figure supplement 1C*), and non-ISNs (*Figure 2—figure supplement 2*)) because none of them shows the second-order paradoxical effects. We therefore concluded that networks with feature-specific EI connectivity can show specific paradoxical effects in response to patterned perturbations, but nonspecific perturbations fail to uncover such effects even in specific ISNs.

## Positive versus negative perturbations of inhibitory neurons

We *decreased* the input to inhibitory neurons in our perturbations, because neurons in strongly connected excitatory-inhibitory networks (similar to those found in the cortex) typically operate at low firing rates. As a result, the paradoxical effect would be observed better if perturbations lead to *increases* in activity of inhibitory neurons to avoid the 'floor effect', which would be possible by *decreasing* the input to inhibitory neurons. To demonstrate this, we probed the presence of the specific paradoxical effect with positive patterned perturbations. We used the same patterns of perturbations, but instead of decreasing the input to inhibitory neurons (negative perturbation) we increased the input (positive perturbation). Positive perturbations with similar size revealed a weak specific paradoxical effect and negative slope (*Figure 2—figure supplement 3A,B*). However, decreasing the size of perturbation by 10 times led to strong negative slopes (*Figure 2—figure supplement 3C,D*), due to less rectification of network activity. Systematic analysis of the slope of the response changes versus input perturbations revealed that smaller perturbations in fact revealed higher negative slopes (*Figure 2—figure supplement 3E*). Negative perturbations, in contrast, always revealed the specific paradoxical effect with strong negative slopes, independent of the strength of perturbation (*Figure 2—figure supplement 3F*). We, therefore, concluded that, although the specific paradoxical effect can be revealed by small positive perturbations, negative patterned perturbations are better poised for probing specific ISNs.

## Networks with more biologically realistic inhibition

We used equal numbers of excitatory and inhibitory neurons with similar tuning properties in our simulations, but it is known that there are fewer inhibitory neurons (than excitatory neurons) in real cortical networks (*Braitenberg and Schüz, 1998*). Moreover, experimental studies suggest broader tuning of inhibitory neurons than excitatory ones (*Ma et al., 2010*; *Bock et al., 2011*). We therefore asked if our main results also hold in networks with such biologically realistic architectures. To address this issue, we first simulated networks with fewer inhibitory neurons, matching the fraction of excitation/inhibition (80% E, 20% I) as reported in the neocortex (*Braitenberg and Schüz, 1998*). To account for the decrease in the overall level of inhibition resulting from the decrease in their number, we increased the average weight of inhibition by a similar factor (4 times stronger), consistent with stronger weights reported for inhibition (*Hofer et al., 2011*). As before, feature-specific paradoxical effect was also observed in such networks, but only when patterned perturbation of inhibition was delivered (*Figure 2—figure supplement 4A*).

We next addressed broader tuning of inhibition. This can either affect the input to neurons (broader input to inhibitory neurons than excitatory ones) or the specificity of connections (more specificity of excitatory compared to inhibitory connections). Different inputs to different neuron types do not change our results, as patterned perturbation is only delivered to inhibitory neurons; this would be an issue if both subtypes were subject to perturbations or external stimuli. We therefore investigated the effect of broader connectivity of inhibitory neurons. To simulate this, we allowed for a broader connectivity of I→E, I→I and E→I connections compared to E→E weights. Moreover, we recalibrated the weights such that the network still has dynamic stability despite the decrease in specific inhibition. For stable networks with broader inhibitory connectivity, we obtained similar feature-specific paradoxical effects in response to patterned perturbations (*Figure 2—figure supplement 4B*). We also combined the two conditions in a third set of simulations, where the network had both fewer inhibitory neurons *and* broader inhibitory connectivity, and obtained similar results (*Figure 2—figure supplement 4C*). Therefore, fewer inhibitory neurons and broader tuning or connectivity of inhibition would not compromise the presence of feature-specific paradoxical effects, as long as the specific neuronal dynamics remain stable.

## Networks with heterogeneous specificity of excitatory and inhibitory connections

We so far considered EI connectivity profiles where all synaptic connections had the same tuning width. However, balanced states and ISN dynamics can be achieved with different profiles of E and I connections. Under such heterogenous connectivity profiles, the relationship between the specific activity dynamics and connectivity might be less clear, as the heterogeneity adds 'noise' which may weaken the specific effect of patterned perturbations. We therefore studied whether specific paradoxical effects can also be observed in neuronal networks with heterogenous specificity of E and I connections (*Figure 2—figure supplement 5*). To this end, we modified the specificity of synaptic connections to have a random value between 0 (non-specific) and 1 (highly specific) (*Figure 2—figure supplement 5A*). Simulation results showed that patterned perturbations of inhibition induced the same inverse pattern of activity on inhibitory and excitatory neurons, but with a more noisy behaviour (*Figure 2—figure supplement 5B*). This reflected itself in a more spread in the distribution of response changes versus input perturbations; however, the overall trend remained the same and manifested itself in the negative slope for patterned perturbations, but not for randomized input patterns (*Figure 2—figure supplement 5C*). The heterogeneity of synaptic connections can, therefore, add more scatter to the manifestation of the specific paradoxical effect, but seems unlikely to mask it in the average slope of the perturbation curve.

## Networks with nonlinear dynamics

As we considered a threshold-linear nonlinearity in our rate-based model, it is possible that our main results were derived mostly in the linear regime of dynamics. More complicated nonlinearities can create interactions amongst the eigenmodes and hence may mitigate the specific paradoxical effects. Specifically, in inhibitory-stabilized networks with supra-linear transfer functions, the dynamic range is divided into non-ISN and ISN regions by a transition from the sub-linear to supra-linear dynamics (*Ahmadian et al., 2013*; *Rubin et al., 2015*). We therefore extended our model to study

how such nonlinear dynamics might affect our results. We investigated the effect of response nonlinearity on our results by simulating rate-based networks with difference expansive nonlinearities (similar to *Ahmadian et al., 2013*) (see Materials and methods). We changed the power of nonlinearity from n = 1 (corresponding to our previous networks, for example in *Figure 2*) to higher powers.

We observed similar feature-specific paradoxical effects in nonlinear networks with different powers of nonlinearity (*Figure 2—figure supplement 6A*). The expansive nonlinearity in fact seems to amplify the specific paradoxical effect, as inhibitory neurons with more negative perturbations show more positive response changes in networks with higher exponents. This effect was concomitant with a higher activation of excitation in networks with higher exponents, which necessitates a more potent specific inhibition in turn (*Figure 2—figure supplement 6B*). Expansive nonlinearity of neuronal transfer function therefore seems to amplify the specific mode in such networks, and as a result the feature-specific paradoxical effects, as long as the dynamics remain stable. In fact, networks with too large nonlinearity (n = 5) showed signatures of such instability in terms of pathological oscillations of excitation and inhibition (not shown). We also tested our results with smaller size of perturbations (up to 10 fold smaller) and found that feature-specific paradoxical effects could still be observed in all nonlinear networks (not shown). Thus, the main signature of feature-specific paradoxical effect can also be observed in networks with nonlinear transfer functions.

## Partial perturbation of inhibition

Previous experimental (*Li et al., 2019*; *Sanzeni, 2019*) and theoretical (*Sadeh et al., 2017*) studies have shown that the fraction of inhibitory neurons perturbed is important to reveal the (non-specific) paradoxical effects in ISNs. We therefore asked whether similar dependence on the proportion of perturbed inhibitory neurons also plays a role in revealing specific paradoxical effects in feature-specific ISNs. Patterned perturbations based on a small fraction of inhibitory neurons in fact failed to reveal any specific paradoxical effect (*Figure 2—figure supplement 7A*). When the patterned perturbation was delivered to half of the inhibitory population the positive regression slope disappeared, but still no sign of feature-specific paradoxical effect (as assayed by a significant negative slope) was present (*Figure 2—figure supplement 7B*). We found that patterned perturbations based on a large fraction of inhibitory population was indeed needed to reveal the feature-specific paradoxical effect, as shown by the negative slope in the example perturbation (70%) (*Figure 2—figure supplement 7C*), and characterized for a larger parameter space (*Figure 2—figure supplement 7D*). We therefore conclude that similar dependence of the paradoxical effect on the fraction of perturbed inhibitory population would be expected also in the case of feature-specific paradoxical effect. However, patterned perturbations can be tuned to be delivered more specifically or more broadly (with non-specific perturbations reflecting one end of the spectrum). It might, therefore, be possible to optimize the experimental protocols, and need a smaller fraction of perturbed inhibition, if partial patterned perturbations are targeted to the most functionally similar units.

## Perturbations based on receptive field similarity to reveal specific inhibitory stabilization

Realistic neurons in visual cortex do not only respond to a single one-dimensional feature like preferred orientation, and the specific connectivity between them is not only a function of their preferred orientation (*Cossell et al., 2015*). Instead, correlations between two-dimensional visual receptive field (RF) of neurons have been reported to be a good predictor of the connection probability and the strength of weights (*Cossell et al., 2015*). We therefore asked whether specific ISN behaviour can be observed in specific EI networks with more realistic RFs and connectivity profiles, resembling cortical networks. To answer this question, we simulated a more realistic version of our model networks, where neurons had Gabor-like RFs (*Figure 3A*), and the specific connectivity between them was modulated by their respective RF correlations (*Figure 3B*; Materials and methods), as suggested by experimental recordings (*Cossell et al., 2015*).

To assess whether specific paradoxical effects are present in the network, we perturbed the input to inhibitory neurons following two protocols: (1) along the 1D dimension of preferred orientation of neurons (similar to *Figure 1*) (*Figure 3C*), and (2) along the dimension of RF similarity (with regards to a reference neuron, see Materials and methods) (*Figure 3D*). The uniform paradoxical effect was present in both perturbations (*Figure 3C1,D1*). However, the specific paradoxical effect was absent

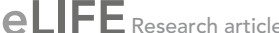

**Figure 3.** Patterned perturbation of inhibition based on receptive field similarity is necessary to reveal specific inhibitory stabilization. (**A**) Sample neuronal RFs for excitatory (magenta) and inhibitory (blue) neurons and schematics of connectivity between neuronal pairs. (**B**) Upper: Distribution of RF similarity of all neuronal pairs in the network, as quantified by pairwise correlations of RFs. Lower: Weights of connections between neurons as a function of their RF similarity. (**C1**) Activity of excitatory and inhibitory neurons in response to patterned perturbation along the 1D dimension of orientation (similar to *Figure 2A*). (**C2**) The average activity of neurons during baseline and perturbed states, along with the profile of inhibitory perturbation, as a function of the preferred orientation of neurons (similar to *Figure 2B*). (**D1,D2**) Similar to (C1,C2) for patterned perturbation along RF similarity (see Materials and methods). The average activity of neurons in (**D2**) is plotted against the RF similarity of respective neurons to a reference inhibitory cell. (**E**) Response change of individual inhibitory neurons as a result of perturbation versus their respective input perturbations, along with the best fitted regression line (red), similar to (**C**). Patterned perturbation along the 1d feature of orientation does not reveal a negative slope, although it yields a nonspecific paradoxical effect (average increase of inhibition as a result of negative perturbations). (**F**) Patterned perturbation along RF similarity (with regard to a reference cell) shows the specific paradoxical effect (negative slope). (**G**) Same pattern of perturbation as in (**F**) but randomized over inhibitory neurons, does not lead to a specific paradoxical effect (lack of negative slope).
The online version of this article includes the following figure supplement(s) for figure 3:

**Figure supplement 1.** Spectral analysis of specific perturbations.

for 1D perturbations (*Figure 3C2*) and was only evident in the perturbations along the RF similarity (*Figure 3D2*). This difference could be quantified by the slope of the regression lines fitted to response change versus input perturbation (*Figure 3E,F*). As a control, randomizing (over inhibitory neurons) the same perturbation protocol as in *Figure 3D* did not lead to the specific paradoxical effect and hence did not reveal specific ISNs (*Figure 3G*). These results suggest that, for realistic neuronal networks, perturbation of the inhibitory population based on their RF similarity might be necessary to reveal specific ISNs, and the reduction of RF properties to 1D features like orientation may not be conclusive in uncovering the specific paradoxical effect.

## Spectral analysis of feature-specific perturbations

Although the perturbation of the network along the 1D dimension of preferred orientation was a *specific* perturbation, it did not lead to a specific paradoxical effect, and hence failed to reveal the

specific ISN nature of the network. To understand why and how this happened, we resorted to theoretical analysis of the network connectivity (*Figure 3—figure supplement 1*). The network connectivity based on RF similarity does not show a strong specific modulation along the dimension of preferred orientation, compared to networks with 1D feature-specific connectivity (*Figure 3—figure supplement 1A,B*). This is due to the fact that neurons with 2d RFs are tuned to multiple features, therefore, even if a pair of neurons are highly similar in their preferred orientation, they might still be different in other respects (e.g. preferred phase or spatial frequency). As a result, feature-specific connectivity based on full RF similarity would be diluted when projected over individual 1D dimensions. Hence, delivering optogenetics perturbations along 1D features might not be strong enough to activate the actual specific mode(s) of the network that would result from its specific connectivity.

To uncover the specific eigenmode of the network along which the perturbations would result in the specific paradoxical effect, we preserved the E-to-E connections as specific as before while randomizing the rest of the connection weights (E-to-I, I-to-E and I-to-I). This procedure preserves the unstable specific excitatory eigenmode of the network, while ensuring that inhibition is strong enough to suppress the unstable nonspecific excitatory mode (this would have otherwise masked the specific E-to-E eigenmodes). The largest, unstable eigenmodes of the connectivity matrix would now be specifically associated with the *specific* modes.

By focusing on the largest eigenvalue in the spectrum of the weight matrix (*Figure 3—figure supplement 1C*) we can now reveal the structure of the specific eigenmode. We found that the corresponding eigenvector had high correlations along the dimension of RF similarity. This is shown for one example cell as reference in *Figure 3—figure supplement 1D*, and for all excitatory cells as different reference RFs in *Figure 3—figure supplement 1E*. In contrast, such a high correlation was not observed when sorting neurons according to their preferred orientation (*Figure 3—figure supplement 1D*). Similar results were obtained when a smaller, but still unstable specific eigenvector was analysed (*Figure 3—figure supplement 1F,G*). These results show that the specific eigenmodes of the weight matrices based on RF similarity of neurons with multiple features may not have strong projections over their individual features. Patterned perturbation protocols based on a single feature like orientation might, therefore, not recruit the specific eigenmode of the network effectively and hence fail to reveal specific paradoxical effects even in the presence of feature-specific ISNs.

## Perturbation patterns based on response similarity to reveal specific ISN

Mapping the RF of inhibitory neurons might be difficult in practice to guide the optogenetics perturbation, especially given that a large population of inhibitory neurons should be imaged and perturbed to reveal specific ISNs. However, experimental studies have shown that both RF similarity and response similarity (namely, how similarly they respond to similar stimuli) are good predictors of the specificity of connections (*Cossell et al., 2015*). We, therefore, explored the possibility that response similarity of neurons, rather than their RF similarity, could be used as a proxy for specific connectivity to guide the patterned perturbation (*Figure 4*).

We stimulated the activity of the network in response to visual stimuli with the same statistics as the ensemble of neuronal RFs (see Materials and methods). The similarity of neuronal responses was then quantified by the correlation of activity traces in response to the entire stimulus set. Overall, neuronal pairs with more RF similarity showed more response similarity, although the match was not perfect (*Figure 4A–C*). However, when we perturbed the inhibitory network in a patterned manner based on response similarity, we observed the specific paradoxical effect (an example is shown in *Figure 4D*). Here, inhibitory neurons with more response similarity (rather than RF similarity as in *Figure 3D*) to a reference cell received more negative perturbations (i.e., *decreased* external input), and inhibitory neurons with more negative perturbations in fact showed *increased* activity in a paradoxical manner.

We repeated this procedure for all inhibitory neurons as reference cells and calculated the slope of the regression line (as described above), as a metric for the presence or absence of specific paradoxical effects. Patterned perturbations based on response similarity to a majority of reference inhibitory cells revealed specific paradoxical effect (*Figure 4E*). We quantified this in terms of the fraction of the inhibitory neurons that would yield the specific paradoxical effect, if the patterned perturbation is based on response similarity of the population in reference to them, and found that perturbations based on >80% of inhibitory neurons as the reference cell in fact lead to a specific paradoxical



**Figure 4.** Patterned perturbation of specific ISNs according to response similarity. (**A**) Left: RF similarity (quantified as pairwise RF correlation) of responsive inhibitory neurons in the network. Right: Response similarity of same inhibitory neurons, calculated as correlation of activity in response to a sequence of stimuli, composed of RFs with similar statistics as the neuronal RFs (see Materials and methods). Responsive units are identified as neurons with average activity more than the 20th percentile of the population and sorted according to their preferred orientation (PO sorted). (**B**) Marginal distribution of response similarity of all neuronal pairs in (**A**). (**C**) Response correlation of the neuronal pairs versus their RF correlation. (**D**) Response change of inhibitory neurons versus their input perturbation, when patterned perturbation is applied according to response similarity with regard to an example reference inhibitory cell. Red line shows the best fitted regression line and the slope is denoted in red. (**E**) The slope of the fitted regression line to the data points (as in (**D**)) for different inhibitory neurons used as the reference. Only significant regression lines (p-value<0.05) have been included. Negative slopes denote specific paradoxical effects. (**F**) Fraction of inhibitory neurons that would reveal a specific paradoxical effect (significant negative slope; red) or not (significant positive slope; black), if used as reference for delivering patterned perturbations.
The online version of this article includes the following figure supplement(s) for figure 4:

**Figure supplement 1.** Patterned perturbation of specific ISNs according to response similarity when full-field gratings are used as stimuli.
**Figure supplement 2.** Feature-specific paradoxical effects obtained by perturbing inhibitory neurons based on their response similarity to natural images.

effect (*Figure 4F*). We, therefore, concluded that it is possible to use response similarity, rather than RF similarity of inhibitory neurons, as a way to deliver patterned inhibition in order to detect specific ISNs.

Interestingly, we found that when full-field oriented gratings were used as visual stimuli, patterned perturbations based on response similarity were not as effective in inducing the specific paradoxical effect (*Figure 4—figure supplement 1*). We used gratings of either fixed spatial frequency (*Figure 4—figure supplement 1A*) or gratings with a range of spatial frequencies (*Figure 4—figure supplement 1D*) to stimulate the network and then calculated the response similarity of neuronal pairs from their responses to the entire stimulus set (*Figure 4—figure supplement 1B,E*). When using grating stimuli with a fixed spatial frequency, most of the reference inhibitory cells used to guide the patterned perturbation of inhibition did not reveal a specific paradoxical effect (*Figure 4—*

*figure supplement 1C*). The fraction of reference cells leading to the specific paradoxical effect was somewhat higher for gratings with a wide range of spatial frequencies (*Figure 4—figure supplement 1F*), but still lower than stimuli with the same statistics as the RFs (c.f. *Figure 4F*).

The reason why such stimuli are not as effective could be due to strong co-activation of cells with similar preferred orientations. Response correlations resulting from such full-field gratings are over-emphasizing the oriented feature of RFs rather than other features, while the specific connectivity (and hence the specific dynamics of the network) is in fact governed by multiple features of RFs. The pattern of perturbation based on such response similarity thus resembles specific perturbations based on 1D features described before (c.f. *Figure 3*), and hence similar arguments in terms of weaker projection of the specific eigenmode over individual features (*Figure 3—figure supplement 1*) might be responsible here too for the failure of such patterned perturbations to invoke the specific paradoxical effect. These results suggest that the choice of stimuli to probe response similarity would be important in designing optogenetic experiments to reveal specific ISNs, with stimuli matching the statistics of RFs having higher statistical power. To test this hypothesis more directly, we used natural images (*Figure 4—figure supplement 2A*) as stimuli to probe the response correlation of neurons in the network (*Figure 4—figure supplement 2B,C*). We found that patterned perturbations based on such response similarity led to specific paradoxical effects for a majority of reference inhibitory cells (*Figure 4—figure supplement 2D,E*). We therefore conclude that naturalistic stimuli, which contain multiple visual features like neuronal RFs, would provide the best protocol to assay response correlations needed for patterned perturbations.

## Specific inhibitory stabilization in spiking networks

So far, we have established that large-scale networks of rate-based model neurons with highly specific EI connectivity behave as feature-specific ISNs, and that this property can be revealed by specific paradoxical effects in response to proper patterns of perturbation. Rate-based models are useful theoretical tools to approximate the average activity of more realistic neuronal networks, and they are especially suited to describe the collective dynamics of such networks (*Destexhe and Sejnowski, 2009*). However, biological neurons communicate with spikes, and spiking networks are known to show richer and more complex activity dynamics (*Brunel, 2000*). Direction demonstration of the presence of specific ISNs, and the associated specific paradoxical effect in them, thus remains to be provided in spiking networks.

To this end, we extended our previous models to simulate large-scale networks of spiking neurons with specific EI connectivity (*Figure 5*; see Materials and methods). We simulated a spiking version of our model (*Figure 5A*), where patterned perturbation was delivered by reducing the baseline input to inhibitory neurons according to their similarity profile. The model showed both uniform and specific paradoxical effects in the average firing rates of neurons (*Figure 5B*). The specific paradoxical effect could be quantified by the negative slope of rate changes versus input perturbations (*Figure 5C*). We also tested whether our results hold in spiking networks with fewer inhibitory neurons and found that this is indeed the case (*Figure 5—figure supplement 1*). These results corroborate that specific paradoxical effects can also be revealed in more realistic spiking networks with specific EI connectivity, if patterned perturbation is delivered to probe that.

## Spontaneous transitions between selective patterns of neural activity

Recent work in mouse visual cortex has suggested that spontaneous activity is not random. Instead, distinct subpopulations of neurons are active together in the absence of direct visual stimulation (*Miller et al., 2014*). These spontaneously emerging neuronal activity patterns are, however, similar to those arising during evoked states (*Miller et al., 2014*). Similar observations had been reported before in other species with structured visual maps (*Kenet et al., 2003*). It is, therefore, not clear which structural or connectivity motifs leads to such intrinsically generated patterns of activity. We next asked if such specific dynamics of spontaneous activity can emerge in networks with highly specific EI connectivity, and whether specific ISNs show similarity between their spontaneous and evoked activity. To study this, we simulated our spiking network models with different levels of connection specificity and probed the network responses to spontaneous and evoked activity (*Figure 6*). Spontaneous activity was emulated by providing unspecific input to all neurons independent of their respective selectivity (i.e., their initial preferred orientations here). Evoked activity, in contrast, was

**Figure 5.** Spiking networks with specific EI connectivity show specific inhibitory stabilization. (**A**) Sample raster plot of spiking activity of excitatory (red) and inhibitory (blue) neurons in the baseline states (gray) and during patterned perturbation (orange). The network has all-to-all connectivity and connection weights are modulated as a function of the preferred orientation of neurons (see Materials and methods). (**B**) Average activity of excitatory and inhibitory neurons during the baseline and perturbed states, across 10 repetitions of the simulation in (A). Input perturbation of inhibitory neurons is shown on the right for comparison. (**C**) Average response change of inhibitory neurons versus their input perturbation. The red line shows the best fitted regression line to the data points.

The online version of this article includes the following figure supplement(s) for figure 5:

**Figure supplement 1.** Specific inhibitory stabilization in spiking networks with fewer inhibitory neurons.

measured in response to tuned inputs with different degrees of specificity (quantified by the fraction of the tuned component of the input to its unspecific part; see Materials and methods).

Networks without specific EI connectivity showed uniform activity patterns in the spontaneous state (*Figure 6A*) and highly selective responses even to weakly tuned inputs in the evoked state (*Figure 6B*). In contrast, networks with specific EI connectivity showed fast transitions between highly selective states (as quantified by the selectivity of population responses; see Materials and methods) (*Figure 6C*), while preserving their selectivity in response to weakly tuned inputs in the evoked state (*Figure 6D*). The fast time scale of spontaneous transitions can be characterized by calculating the correlations between selective patterns of the excitatory population as a function of the time interval between their occurrences (*Figure 6—figure supplement 1*). Networks with balanced specific EI connections, where the specificity of EE connections was matched by the specificity of other

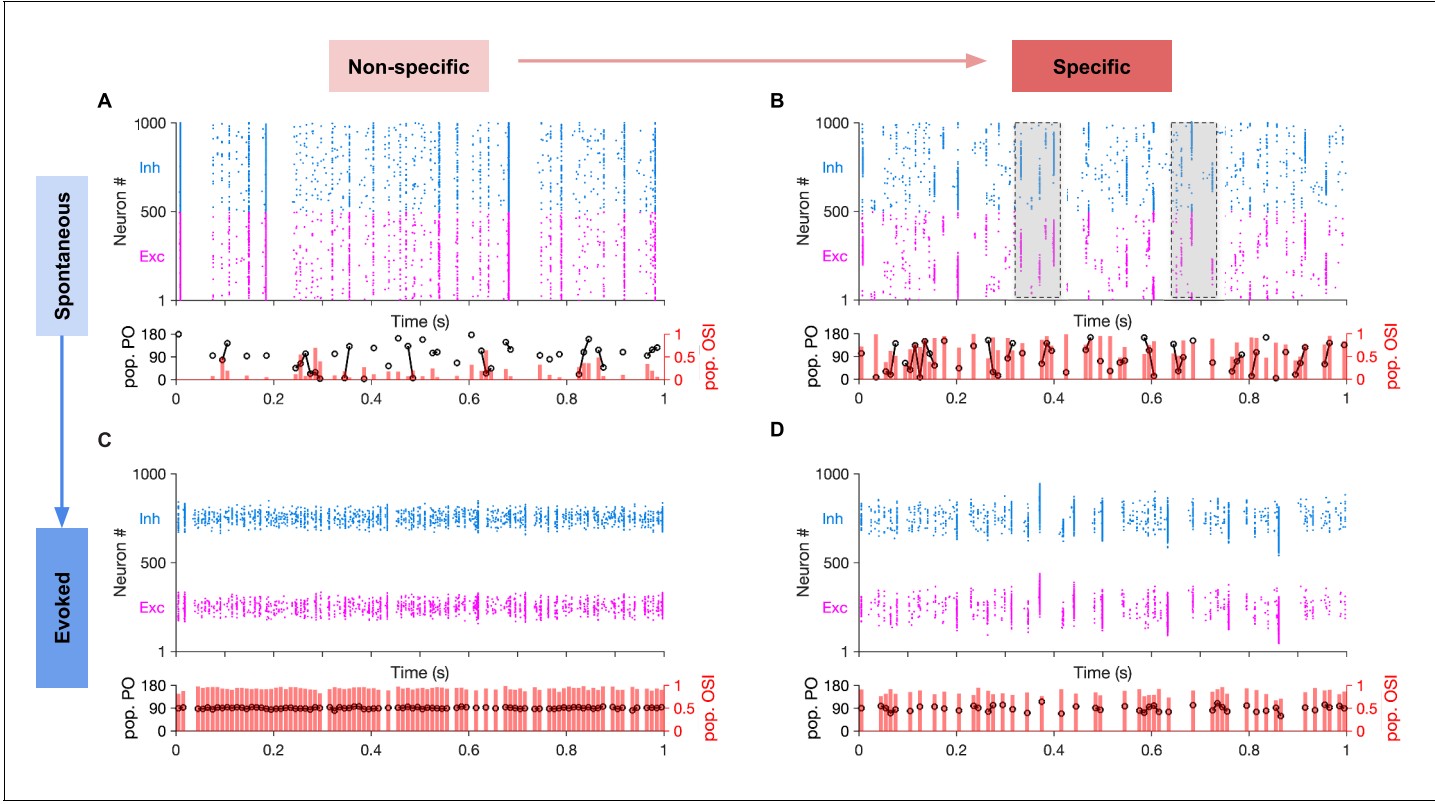

**Figure 6.** Spontaneous transition between selective states during spontaneous activity in specific EI networks. (**A**) Sample raster plots of spontaneous spiking activity of excitatory and inhibitory neurons for network with nonspecific EI connectivity. Neurons are sorted according to their preferred orientation (from 0 to 180 degrees). Population preferred orientation (pop. PO, black) and population orientation selectivity index (pop. OSI, red) are shown at the bottom, for the activity of the excitatory population rendered in bins of 10 ms (Materials and methods). Only bins in which at least five excitatory neurons are active are included in the analysis. (**B**) Same as (**A**) for the spontaneous activity of networks with specific EI connectivity (Specific connectivity: 100%, corresponding to m = 1; see Materials and methods). Neurons show clustered activity around different preferred orientations, in the absence of a tuned external input with a specific orientation. The transition between highly selective states with different neuronal patterns (clustering of activity around different POs) is highlighted for two example regions (shaded). Note population activity patterns with different pop. POs and high pop. OSI in the vicinity of each other. (**C,D**) Same as (**A,B**), respectively, for evoked activity. During evoked activity, stimulus with a specific orientation (90°) is shown, conveying tuned input (Specific input: 100%) to neurons according to their preferred orientations. See Materials and methods for details. The online version of this article includes the following figure supplement(s) for figure 6:

**Figure supplement 1.** Time scale of transitions between selective states during spontaneous activity.

connections, showed a very fast fall off of pattern correlations (*Figure 6—figure supplement 1A*). In contrast, networks with broader inhibition exhibited very long time scales of pattern correlations (*Figure 6—figure supplement 1B*). These results demonstrate that, although networks with highly specific excitatory connectivity can show selective spontaneous activity, fast transitions between these selective states only appear in networks that are balanced in EI connection specificity.

These different behaviours were better illustrated by plotting the distribution of the preferred orientations (PO) of temporary population responses for each of the four examples (*Figure 7A*). Transitions between highly selective population responses would translate to a wide range of population POs with high selectivity (quantified by the orientation selectivity index, OSI, of the population). Nonspecific networks in the spontaneous activity showed a wide range of population POs, but highly selective population responses were rare, indicating that non-specific networks predominantly remain in the same, non-specific state. In contrast, specific networks showed spontaneous population responses with a wide range of POs and high OSIs at the same time (*Figure 7A*, upper), consistent with spontaneous transitions between highly selective ensembles. Both networks, however, showed population responses with POs close to the stimulus orientation (90° here) when stimulated with weakly tuned inputs (*Figure 7A*, lower), although the specific network still showed more



**Figure 7.** Quantification of spontaneous transition in networks with nonspecific and specific EI connections. (A) Distribution of population POs derived from the four raster plots shown in *Figure 6*, for all the data (black) and for highly selective (orange) moments of population activity (pop. OSI > 0.5). (B) The spontaneous transition (ST) index (Materials and methods) for all four combinations of spontaneous (Spont.) and evoked activity of networks with nonspecific (NS) or specific (SC) EI connectivity. Red bars show the mean and std of shuffled data for each case (see Materials and methods for details). (C) ST index for responses of networks with different degrees of specific connectivity (Spec. Conn.) and in response to inputs with different levels of specificity (Spec. Inp.). All results in this figure are obtained from similar networks as in *Figures 5* and *6*, but with longer simulation times (10s).

variability in the preferred orientation of the population activity. This variability is reminiscent of recurrently generated noise in clustered networks, which is suppressed by stimulus presentation (*Litwin-Kumar and Doiron, 2012*).

To quantify transitions between selective states, we developed a spontaneous transition (ST) index which combines both aspects of transition (showing a wide range of population POs) and selectivity (showing highly selective population responses) into a single metric (see Materials and methods). Higher ST values indicate higher frequency of visiting different and selective states, while low values indicate either lack of transition or transition between nonspecific states of neural activity. The ST index was very high for spontaneous responses of networks with specific EI connectivity but remained low for all other combinations (*Figure 7B*). Development of the ST index to quantify spontaneous transition in each network allowed us to explore the parameter space more systematically. We simulated spiking networks with various degrees of specific connectivity in response to inputs with different levels of tuning and calculated the ST index for each network. The result revealed a landscape where selective spontaneous transitions are observed for networks with highly specific EI connectivity. The spontaneous transitions are, however, quenched in response to tuned inputs carrying stimulus information in the evoked state (*Figure 7C*). We therefore conclude that spiking networks with highly specific EI connectivity can show rapid transitions between selective states during spontaneous activity, and this is consistent with their selective stimulus-evoked responses to tuned inputs.

To directly test whether there is a relationship between the dynamical signature of specific ISNs (namely, the feature-specific paradoxical effect) and spontaneous transitions between selective states, we quantified both in spiking networks with different degrees of specific connectivity (*Figure 8*). In each network, we calculated the ST index in the spontaneous activity and the slope of the inhibitory response changes as a function of patterned perturbations (sample networks for low, medium and high specific connectivity in *Figure 8A,B*). Comparing the ST index with the slope for different networks revealed that there is indeed a correlation between negative slopes (i.e., feature-specific paradoxical effect) and higher spontaneous transitions (*Figure 8C,D*). Transition to feature-specific ISNs was in fact correlated with a rapid jump in the ST index from low values (*Figure 8D*), indicating a change in the regime of activity. These results show that emergence of spontaneous transitions between selective states and the presence of specific paradoxical effects (as a dynamical signature of feature-specific inhibitory stabilization) are concomitant effects in networks with specific EI connectivity.

## Discussion

By simulating computational models at different levels of realism, and through theoretical analysis of such networks, we showed here that specific inhibitory stabilized networks (specific ISNs, where unstable specific excitatory connectivity is balanced by specific inhibition) can show 'specific paradoxical effects'. We further showed that, to reveal such specific ISNs, patterned perturbation of inhibition is needed. Our study thus extends classical inhibitory-stabilized networks, where only the uniform paradoxical effect was studied in response to nonspecific perturbations of inhibition. Our results suggest that specific ISNs show specific paradoxical effects *in addition* to the uniform paradoxical effect, provided they are probed by the proper patterns of perturbation, which would effectively stimulate the specific dynamics of the network. Absent such patterned perturbation, they would behave the same as nonspecific ISNs and hence their specific inhibitory stabilization would remain undetected.

Operation of cortical networks in specific ISN regimes adds another dimension to the computational capacity of the brain. It argues for the presence of specific directions of amplification of the feedforward input by the cortical dynamics, which is stabilized by a precisely tuned inhibition along the same direction. Our study suggests that, if such specific computational dimensions existed or emerged in cortical networks, they would be manifested by the specific paradoxical effects in response to patterned perturbations. While the classic paradoxical effect is observed by perturbing and measuring the average activity of inhibitory neurons in a non-specific manner (e.g. *increased* average activity when the perturbation *decreases* the average input), the specific paradoxical effect is observed in the relation between the residual input and output response of inhibitory neurons around the mean. We provided a quantitative metric for measuring this specific paradoxical effect, by evaluating the slope of the regression line fitted to response changes as a function of input perturbations (e.g., *Figure 2C*). Note that, as such, the measure would be robust to different rate changes that might be induced for neurons with different firing rates (e.g., cells with high firing rates will have their absolute rate decreased more than a cell firing sparsely in response to external inhibitory current injection). Such a dependence would not change the sign of the slope, which is important to infer the specific paradoxical effect.

In reality, it is difficult to measure connectivity for every individual circuit before optogenetically perturbing it. We therefore used feature or RF specificity as a proxy for connection weights, since it has been shown to be a good predictor of connectivity (*Cossell et al., 2015*). Our results, however, indicated that perturbation patterns based on one-dimensional features of neurons, for example their orientation selectivity, might not be proper protocols to probe specific ISNs as they would not effectively recruit the specific dynamics of the network. Considering the full RF similarity induced a better and more effective way of unravelling specific ISNs, and our spectral analysis revealed the underlying difference between the two perturbation protocols. Based on these results, we predict that experimental protocols relying on a single one-dimensional feature to deliver patterned perturbations may not be as potent to reveal specific ISNs, compared to patterned perturbations based on full RF similarity.

Receptive field mapping can, however, be equally costly, especially given that large-scale networks of inhibitory neurons need to be imaged and perturbed to see the specific ISN effects. We



**Figure 8.** Correlation between specific paradoxical effects and spontaneous transitions between selective states. (A) Sample raster plots of spiking activity from networks with different degree of specific EI connectivity (m = 0, 10, 100%, respectively). Networks and conventions are otherwise similar to *Figure 6*. (B) Normalized response change of inhibitory neurons as a function of the normalized change in their input due to patterned perturbations. Response changes (Resp. change) and input perturbations (Inp. perturb.) are normalized to the average firing rate and the average input in the baseline state, respectively. The network is simulated for 5s in the baseline state, and another 5s during patterned perturbation. Numbers in red denote the slope of the best fitted regression line to the data (red line). The procedure and conventions are otherwise similar to *Figure 5C*. (C) The ST index (as in *Figure 7*) as a function of the slope of perturbations for different networks (back dots). The mean and std of the ST index for shuffled activity (100 repetitions) are shown for comparison (grey). (D) The bootstrapped ST index for networks with negative (specific paradoxical effect) and positive (normal) slopes of perturbations. (E) Bootstrapped ST index and the normalized specific ISN effect (Spec. ISN) in networks as a function of feature-

*Figure 8 continued on next page*

*Figure 8 continued*
specific connectivity (m). Spec. ISN for each network is calculated by normalizing the slope of the perturbation by slope in the network with the maximum specific connectivity (m = 100%).

therefore explored the alternative possibility of using response similarity (assayed by how similarly neurons respond to an ensemble of stimuli) to deliver patterned perturbations, since response similarity has also been suggested as a good predictor of network connectivity (*Cossell et al., 2015*). Patterned perturbation based on response similarity would be possible by all-optical techniques which enable reading and writing the neuronal activity at the same time (*Emiliani et al., 2015*; *Zhang et al., 2018*). Our results revealed that specific paradoxical effects can be induced with such perturbation protocols, but the choice of the ensemble of stimuli used to probe the network responses could be crucial. Stimuli with statistics similar to RFs were more powerful in revealing specific ISNs, while wide-field gratings with fixed or diverse spatial frequencies did not prove to be as effective. While we generated stimuli with the same statistics as neuronal RFs in our simulations, we expect that natural images lead to the same results. Based on these results, we therefore suggest that the most effective and feasible protocol to detect specific ISNs would be to perturb the neuronal networks based on their response similarity in response to natural images, with read/write optogenetics tools.

In order to reveal the specific paradoxical effect, patterned perturbations can be induced by either decreasing or increasing the input to inhibitory neurons (positive or negative perturbations, respectively). Negative perturbations, however, were proven to be more powerful in our study (*Figure 2—figure supplement 3*). This is due to the fact that the 'floor effect' limits positive perturbations: in sparsely active networks (like the visual cortex), potent activation of inhibitory neurons can lead to strong rectification of the network activity. Such intensity-dependent optogenetics effects were in fact implicated as a source of conflict in the results of previous studies on the functional role of different subtypes of inhibitory neurons (*Lee et al., 2012*; *Wilson et al., 2012*; *El-Boustani et al., 2014*). If excitatory neurons are already active at very low rates due to balance of excitation and inhibition, strong positive perturbations of inhibition would lead to silencing of the network, which can obscure the paradoxical effect by decreasing the 'effective' size of the network involved in excitatory-inhibitory interaction. Decreasing the input to inhibitory neurons, on the other hand, increases the activity of excitatory neurons arbitrarily. Although the specific paradoxical effect can in principle be revealed with infinitesimally small positive perturbations, in realistic experimental conditions with significant measurement noise, such small perturbations would decrease the signal-to-raise ratio, hence limiting the statistical power to infer the presence of the negative slope of response changes versus input perturbations. Our results thus suggest that experimental protocols relying on negative patterned perturbations would be more powerful to reveal specific ISNs.

**Table 1.** Parameters of rate-based networks.

| Description | Type | Symbol | Figure 2 | Figure 2—figure supplement 2 | Figure 2—figure supplement 1 | Figure 3, 4 |
|---|---|---|---|---|---|---|
| Number of neurons | E | $N_E$ | 400 | 400 | 400 | 400 |
| | I | $N_I$ | 400 | 400 | 400 | 400 |
| Time constant | E and I | | 10 | 10 | 10 | 10 |
| Synaptic weight | E→E | $J_{EE}$ | 0.05 | 0.001 | 0.05 | 0.05 |
| | E→I | $J_{IE}$ | 0.05 | 0.001 | 0.05 | 0.05 |
| | I→E | $J_{EI}$ | −0.075 | −0.0015 | −0.075 | −0.075 |
| | I→I | $J_{II}$ | −0.075 | −0.0015 | −0.075 | −0.075 |
| Synaptic specificity | E→E | $m_{EE}$ | 1 | 1 | 0 | 0.5 |
| | E→I | $m_{IE}$ | 1 | 1 | 0 | 0.5 |
| | I→E | $m_{EI}$ | 1 | 1 | 0 | 0.5 |
| | I→I | $m_{II}$ | 1 | 1 | 0 | 0.5 |
| Specificity exponent | | η | - | - | - | 2 |

**Table 2.** Parameters of spiking networks.

| Description | Type | Symbol | Figure 5 | Figures 6–8 |
|---|---|---|---|---|
| Number of neurons | E | $N_E$ | 500 | 500 |
| | I | $N_I$ | 500 | 500 |
| Membrane time constant | E and I | $\tau_m$ (ms) | 20 | 20 |
| Threshold voltage | E and I | $V_{th}$ (mV) | 20 | 20 |
| Reset voltage | E and I | $V_{reset}$ (mV) | 0 | 0 |
| Synaptic weight | E→E | $J_{EE}$ (mV) | 2 | 2 |
| | E→I | $J_{IE}$ (mV) | 2 | 2 |
| | I→E | $J_{EI}$ (mV) | -4 | -4 |
| | I→I | $J_{II}$ (mV) | -4 | -4 |
| Synaptic specificity | E→E | $m_{EE}$ | 1 | [0 … 1] |
| | E→I | $m_{IE}$ | 1 | [0 … 1] |
| | I→E | $m_{EI}$ | 1 | [0 … 1] |
| | I→I | $m_{II}$ | 1 | [0 … 1] |

Another experimental constraint for probing the presence of specific ISNs in the cortex is related to the span of recording fields. As patterned perturbations are delivered with regard to functional properties of neurons (e.g. their visual RFs), the more similar the neurons are, the more difficult it becomes to assess the slope of their response changes to input perturbations. Therefore, if the window of imaging is too small and/or perturbations are delivered too locally, the visual RFs might become too similar, which would limit the range of input perturbations based on RF dissimilarity (x-axis in *Figure 2C*). This, in turn, impedes the assessment of the specific paradoxical effect, by limiting the statistical power of measurements to infer a significant negative slope. Wide-field perturbation/imaging is, therefore, necessary to obtain enough heterogeneity in RFs of inhibitory neurons needed for such assessment.

Our study here has also important implications for neuronal connectivity in the cortex. We argued that networks with strong specific excitatory connections, as reported in the visual cortex of rodents (*Ko et al., 2013*; *Cossell et al., 2015*), would be unstable in the absence of specific inhibitory stabilization. Although recent experimental findings (*Znamenskiy et al., 2018*) have corroborated this possibility, assessing the presence or absence of specific ISNs in the cortex can be considered as a crucial experiment to investigate the functional specificity of inhibitory connectivity. This may, in turn, shed light on the long-lasting debate over the tuning of inhibitory neurons and the specificity of their connections (*Liu et al., 2011*; *Wilson et al., 2012*; *Runyan and Sur, 2013*). Our results thus offer detailed predictions regarding the observation of specific inhibitory stabilization in cortical networks with specific EI connectivity, while highlighting important constraints on the design of experimental protocols to effectively probe them.

To unravel their dynamical and functional properties, cortical circuits have so far been mainly probed by simple perturbation protocols. Even when optogenetic perturbations are delivered specifically to certain neuronal subtypes, for example parvalbumin positive (PV+) or somatostatin-positive (SOM+) inhibitory cells, they typically stimulate the whole, or a random fraction of the population nonspecifically. Recent advances in optogenetic techniques are enabling us - at an unprecedented pace - to design perturbation protocols where a specific pattern (e.g. a specific sequence or spatiotemporal pattern of activity) is invoked in identified neurons (*Emiliani et al., 2015*; *Ronzitti et al., 2017*; *Chernov et al., 2018*; *Zhang et al., 2018*; *Carrillo-Reid et al., 2019*; *Marshel et al., 2019*; *Russell, 2019*; *Tran et al., 2019*). Our study here proposes that such technological advances are indeed necessary to reveal fundamental principles of cortical computation. Our results in fact suggest a further level of specificity (amplitude) as a prerequisite of patterned perturbation, since not only specific neurons should be perturbed to reveal specific ISNs, but different levels of perturbation (according to specific patterns) should be delivered to achieve effective recruitment of the specific mode. Such amplitude-specific patterns of perturbation, with different levels of perturbation at identified neurons, might be instrumental in effective interrogation of

cortical networks during healthy and unhealthy function (e.g. in detecting specific epileptic modes of activity).

Patterned perturbations can also shed light on the controversy over the nature of the paradoxical effects. While the observation of paradoxical effects in several studies and across different cortices (*Adesnik, 2017*; *Kato et al., 2017*; *Li et al., 2019*; *Sanzeni, 2019*) has been mainly attributed to ISN dynamics, one needs to rule out the possibility of disinhibition underlying this effect, given the complex connectivity of multiple subtypes of inhibitory neurons (*Pfeffer et al., 2013*). A recent study has in fact suggested that cortical circuits can exhibit paradoxical effects because of disinhibitory loops rather than inhibition-stabilization mechanisms (*Mahrach et al., 2020*). Patterned perturbations of the type we have proposed in this work has the potential to distinguish between these two possibilities, for the following reason. By assuming a degree of specific connectivity between pyramidal cells (PC) and PV+ interneurons (*Znamenskiy et al., 2018*), specific inhibitory perturbations delivered to PV+ neurons would reveal the feature-specific ISN dynamics arising within PC-PV circuitry. Disinhibition, namely inhibition of PV+ interneurons by SOM+ interneurons, can only arise from PC→SOM→PV stream (*Pfeffer et al., 2013*). For disinhibition to be feature-specific, we need to have similar levels of specific connectivity not only between PC→SOM, but also SOM→PV connections. Although SOM+ cells have been reported to have higher orientation selectivity than PV+ neurons (*Ma et al., 2010*), they are known to be involved in larger scales of spatial integration, as opposed to local processing of PV+ cells (*Adesnik et al., 2012*; *El-Boustani and Sur, 2014*). It is, therefore, likely that they have larger RFs and mediating feature-specific surround suppression, more than being involved in local processing of visual information, that shapes the type of specific connectivity we explored here. One way of testing this experimentally would be to change the spatial extent of patterned perturbations from local to global and assay the paradoxical effect. This should shed light on whether a disinhibitory mechanism governed by SOM+ neurons is involved in the paradoxical effect. However, mapping the feature-specific connectivity of SOM+ cells with pyramidal cells would ultimately be needed to constrain the computational models for the role of different subtypes of inhibition in feature-specific ISNs.

Our study of specific ISNs revealed different functional properties of these networks. We found that networks with strong specific EI connectivity can undergo spontaneous transitions between highly selective ensembles, reminiscent of what has recently been observed in mouse visual cortex (*Miller et al., 2014*), and similar to what had previously been reported in other species with structured cortical maps (*Kenet et al., 2003*). Our modelling results showed that such fast spontaneous transitions only happen in specific networks and in the absence of tuned input to neurons. Note that, in comparison, classical attractor networks show very slow dynamics (*Goldberg et al., 2004*), and networks with only EE specific connectivity were incapable of reproducing such fast time scales of transition (*Figure 6—figure supplement 1*; *Sadeh et al., 2015b*). This would, in turn, render such models in their simple form inconsistent with the finding that sensory-evoked activity decays rapidly in recurrent circuits after silencing the feedforward input in visual cortex (*Reinhold et al., 2015*). Specific ISNs, on the other hand, are equipped with a mechanism for such fast transitions due to specific EI balance (*Figure 6—figure supplement 1*) and, therefore, highly specific EE connectivity in these networks can be consistent with the experimentally reported time scales.

Rapid transitions between specific states, without being trapped in each attractor for long, can give the network the capacity of revisiting the functional subnetworks spontaneously. This can endow the network with intrinsic mechanism to reactivate the learned states to avoid forgetting, in the absence of specific stimulation, and it might be important for plasticity, such that reactivation during the evoked state does not lead to unlearning (*Litwin-Kumar and Doiron, 2014*). It can also provide a template of most likely stimulus states, thus leading to a more efficient representation of the environment and more parsimonious responses to external stimuli. Other functional benefits of specific ISNs can be in line with the computational advantages envisaged for conventional ISNs, like selective amplification, noise rejection and pattern completion. The specific ISN regime may enable the network to perform these computations specifically along the 'right' dynamics. In fact, the emergence of specific connectivity in developing visual cortex is correlated with an increase in the reliability of responses (*Ko et al., 2013*). Future studies are needed to investigate the contribution of different motifs of connectivity, including specific EI interaction, to this process.

It is also interesting to speculate on the possible mechanisms underlying the emergence of feature-specific ISNs. To obtain feature-specific connectivity of inhibition which specifically balances

excitatory subnetworks, detailed balance as opposed to general balance is needed (*Hennequin et al., 2014*; *Hennequin et al., 2017*). One possibility to achieve such detailed balance is via inhibitory plasticity mechanisms, as suggested for instance by plasticity of I-to-E synapses (*Vogels et al., 2011*). This is in fact consistent with the specificity of I-to-E weights reported in the visual cortex (*Znamenskiy et al., 2018*). However, E-to-I synapses have also been reported to have a significant level of specificity (*Znamenskiy et al., 2018*). A large-scale model of visual processing in balanced networks equipped with a voltage- and spike-dependent plasticity rule (*Clopath et al., 2010*) reported the emergence of a bidirectional specificity of E-I connections (*Sadeh et al., 2015a*). In future studies, it would be interesting to selectivity perturb different connectivity motifs (E-to-I and I-to-E) and plasticity mechanisms and explore the role of each in the emergence of feature-specific ISNs.

In summary, our study of networks with strong and specific excitatory-inhibitory connectivity reveals new dynamical and functional properties arising from specific inhibitory stabilization. Our results suggest that specific ISNs not only show the classical paradoxical effect, reflected in the average activity of the population, but also manifest specific paradoxical effects, which is represented in the pattern of response change of inhibitory neurons in response to external perturbations. To reveal the specific paradoxical effect, however, patterned perturbations aligned with the specific mode of inhibitory stabilization in the network is necessary.

## Materials and methods

### Network modelling

#### Rate-based simulations

We simulated networks of rate-based neurons with dynamics described as:

$$\tau\frac{dr_E}{dt} = -r_E + [W_{EE}\, r_E\, +\, W_{EI}\, r_I\, +\, s_E]_+$$
$$\tau\frac{dr_I}{dt} = -r_I + [W_{IE}\, r_E\, +\, W_{II}\, r_I\, +\, s_I]_+$$

(1)

Here, $r_E$ and $r_I$ are the vectors of firing rates of $N_E$ excitatory (E) and $N_I$ inhibitory (I) neurons, respectively. $W$ is the matrix of connection weights, including connections between E-to-E ($W_{EE}$), E-to-I ($W_{IE}$), I-to-E ($W_{EI}$), and I-to-I ($W_{II}$) neurons. $s$ is the external input, with $s_E$ and $s_I$ denoting inputs to E and I neurons, respectively. $\tau$ is the effective time constant of the network, with a default value of $\tau = 10$ in our simulations. $[]_+$ denotes half-wave rectification. We used forward Euler method to solve for the firing rates of neurons.

To simulate networks with expansive nonlinearity, we solved a more generalized dynamics, described by $\tau dr/dt = -r + ([Wr\, +\, s]_+)^n$, where n determines the power of nonlinearity.

#### Spiking network simulations

Spiking networks were modelled by simulating the equations describing the membrane potential dynamics of leaky integrate-and-fire neurons:

$$\tau_m \frac{dV_m}{dt} = -V_m(t) + s(t)$$

(2)

Here, $V_m$ denotes the membrane potential of a neuron, and $\tau_m = RC$ is the time constant of integration of the membrane potential, where $R$ and $C$ are the membrane resistance and capacitance, respectively. Spiking mechanism is implemented as follows: when the membrane potential reaches a voltage threshold, $V_{th} = 20\, mV$, a spike is elicited and the membrane potential is reset to the reset voltage, $V_{reset} = 0$. We used exact integration method (*Rotter and Diesmann, 1999*) to solve for the membrane potential and spiking activity of neurons.

The momentary input to the neuron is described by $s(t) = R\, I(t)$, which arises from incoming spikes to the neuron and comprises two components: external input from feedforward and background sources (reflecting stimulus-induced input and input from non-local sources, respectively), and internally generated input from recurrent activity. Once a spike is emitted in a presynaptic source, an instantaneous change in the membrane potential of all postsynaptic sources is emulated in the next simulation time step, by the value of $J$ mV. Note that $J$ is already expressed in units of

volts and not current, and hence describes the effect of $RI$ simultaneously. The total input at time $t$ for a postsynaptic neuron $i$ can therefore be written as $s(t) = \Sigma_j J_{ij}\delta_j(t)$, where $\delta_j(t)$ denotes the presence (1) or absence (0) of spike in all presynaptic sources (including external input or input from presynaptic neurons in the recurrent network), and $J_{ij}$ describes the weight of connection (in mV) from the $j$-th presynaptic source.

## Neuronal receptive fields

The identity and stimulus preference of neurons is assigned in two ways:

1. Based on 1D preferred features (preferred orientation as in *Figure 2*), where neurons are assigned a random initial preferred orientation between $[0, \pi]$.
2. Based on 2D visual receptive fields (RFs, as in *Figure 3*), which are assigned by generating a population of 2D Gabor fields with the following parameters:

$$g_{\lambda,\theta,\phi,\sigma,\gamma}(x,y) = \exp\left(-\frac{x'^2 + \gamma y'^2}{2\sigma^2}\right)\cos\left(2\pi\frac{x'}{\lambda} + \phi\right),\tag{3}$$

where

$$\begin{aligned} x' &= x\cos(\theta) + y\sin(\theta)\\ y' &= -x\sin(\theta) + y\cos(\theta). \end{aligned}\tag{4}$$

Here, $x$ and $y$ specify the position on the 2D visual field, and $\theta$ is the preferred orientation of the RF. The standard deviation $\sigma$ determines the size and $\gamma$ controls the aspect ratio of the receptive field, the parameter $\lambda$ determines the wavelength (with $1/\lambda$ determining the spatial frequency, $\omega$), and $\phi$ is the spatial phase of the field.

Default parameters are chosen as follows: $\theta$ and $\phi$ are drawn randomly from a uniform distribution between $[0, \pi]$, $\sigma = 2.5$, $\gamma = 0.5$, spatial frequency ($\omega = 1/\lambda$) is drawn from a gamma distribution with shape parameter 2 and scale parameter 0.04. RFs are instantiated in a $L \times L$ field, where $L = 50$ specifies the extent of the visual field in each direction in degrees. The resolution of the instantiated fields was 4 pixels per degree (ppd). Each RF was centreed at a specific position, $(x_0, y_0)$, which was close to the centre of the field, $(0, 0)$, but with some random variation, $(\delta x, \delta y)$, which were in turn drawn randomly from $[-1.25, 1.25]$ degrees.

## Network connectivity

Network connectivity is implemented similarly in rate-based and spiking networks, with the entry of the weight matrix $W$ corresponding to presynaptic neuron $j$ and postsynaptic neuron $i$ ($w_{ij}$) denoting the strength of connections in both. This quantity is in arbitrary units for rate-based networks, and it is expressed in mV for spiking networks. In both networks, all neurons are connected to each other (all-to-all connectivity) and the specific connectivity is obtained by modulating the connection weights based on the similarity of pre- and postsynaptic neurons. The similarity of neuronal pairs is defined based on their RF type:

For neurons with RFs based on 1D features (e.g. orientation selective neurons in *Figure 2*), the weight matrix is constructed as 1D feature-specific (e.g. orientation-specific) connectivity. Connection weights between neurons $i$ and $j$ are modulated according to the difference of their preferred orientation as the following:

$$w_{ij} = J\left(1 + m\cos\left[2\left(\theta_i^* - \theta_j^*\right)\right]\right)\tag{5}$$

where $\theta^*$ is the preferred orientation in radians, and $m$ specifies the degree of specific connectivity (ranging from 0%, for nonspecific, to 100%, for maximum specific connectivity). To add randomness, each weight is multiplied by $\zeta$, which is a random number between 0 and 2. Note that this procedure does not change the mean weights, while randomly amplifying or attenuating the deterministic weights between 0 to 100%.

For neurons with 2D visual RFs (e.g. as in *Figure 3*), the specific connectivity was constructed based on RF similarity of neuronal pairs. Connection weights between neuronal pairs are modulated as a function of their respective RF correlations according to:

$$w_{ij} = J\left(0.1 + m\,\exp\left(\eta\,\psi_{ij}\right)\right) \tag{6}$$

where $\psi_{ij}$ is the pairwise correlation coefficient of visual RFs of neurons $i$ and $j$, $m$ denotes the degree of specific connectivity as before, and $\eta$ determines the sharpness of the exponential dependence of weights on RF correlations (see e.g. *Figure 3B* with $\eta = 2$).

## Network stimulation and perturbation

During baseline state, all neurons in the networks received similar levels of input from external sources. This was emulated as a constant input with some random variation for rate-based models or a sequence of spikes generated from a Poisson process with a fixed rate for spiking simulations. Upon perturbation, the input to inhibitory neurons was *reduced*, either by decreasing the input (in rate-based simulations) or decreasing the rate of incoming spikes (in spike-based models) to inhibitory neurons. Perturbations were either performed randomly, by ignoring the identity of inhibitory neurons and decreasing the input to them with a random value generated from a uniform distribution, or according to a pattern respecting neuronal identity (e.g. as in *Figure 2B*, lower). To control for the distribution of perturbations, we also randomized the same pattern over neurons, and used it as a randomized control for perturbations (c.f. *Figure 2B*, upper).

Patterned perturbations according to 1D feature selectivity (e.g. in *Figure 2*) was delivered by changing the baseline input to inhibitory neurons with respect to their orientation selectivity:

$$\delta s_I = -\gamma\left[\sin\left(2\theta_I^*\right) - 1\right] \tag{7}$$

Here, $\delta s_I$ is the vector of perturbation of inhibitory neurons, $\theta_I^*$ is the vector of preferred orientation of inhibitory neurons, and $\gamma$ denotes the size of perturbation. For networks containing neurons with 2D RFs (e.g. in *Figure 3*), patterned perturbation was delivered with regard to RF similarity of inhibitory neurons to a reference inhibitory neuron, $k$, as follows:

$$\delta s_I = -\gamma\left[\exp\left(\psi_I^k\right)\right] \tag{8}$$

where $\psi_I^k$ represents the vector of RF correlation of inhibitory neurons to the reference cell, and $\gamma$ denotes the size of perturbation as before. We also delivered patterned perturbations based on response similarity for these networks (*Figure 4*), where $\psi_I^k$ was replaced by the response correlation of neurons in response to a sequence of stimuli, as described below.

To probe response similarity or neurons in the network, we simulated the networks (as in *Figure 4*, and *Figure 4—figure supplement 1*) with four different stimulus sets. First, stimuli were generated following the same procedure used to generate 2D Gabor fields for neuronal RFs, with the same parameters as described above. Therefore, RFs and stimuli had the same statistics although they were obviously different. Two other sets of stimuli consisted of full-field gratings (examples shown in *Figure 4—figure supplement 1*) with different orientations and spatial phases, both randomly drawn from the range $[0, \pi]$. One set has a fixed spatial frequency of $\omega = 0.04$, while the other had a wide range of spatial frequencies matching the distribution of spatial frequencies of RFs. The fourth set of stimuli consisted of natural images, which were taken from the McGill calibrated image database (http://tabby.vision.mcgill.ca). The central part of each image was chosen, with the same size (in pixels) as RFs (that is, the central 200x200 pixels).

For each stimulus type, a sequence of 200 stimuli was presented to the network. Upon presentation of each stimulus (for 200 ms), the external input to each neuron was modulated by the correlation of its RF with the stimulus field, according to $s = 1 + \beta\psi_{ij}^{st}$, where $\psi_{ij}^{st}$ denotes the correlation of the i-th stimulus field with the j-th RF, and $\beta$ specifies the depth of modulation. The activity trace of neurons in response to the full sequence of stimuli was then correlated with each other to generate a measure of response similarity. The parameters of *Equation 8* were chosen to be $\gamma = 0.25$ and $\kappa = 2$ for all four stimulus types. $\beta = 1$ for full-field gratings and natural images (*Figure 4—figure supplement 1* and *Figure 4—figure supplement 2*), but we chose a smaller value for RF-like stimuli ($\beta = 0.5$), as they had higher correlations with RFs on average (*Figure 4*).

Other parameters of network simulations are provided in *Table 1* and *Table 2*, for rate-based and spiking networks, respectively.

## Data analysis

To quantify the specific paradoxical effect (e.g., as in **Figure 2C**), we fitted lines to response changes (obtained as the difference of the average activity during perturbation and baseline states, after discarding the initial transient period) versus respective input perturbations. The slope and the significance of the best fitted linear regression was used, with significantly negative slopes denoting specific paradoxical effects.

Preferred orientation (PO) and orientation selectivity index (OSI) of population responses in **Figure 6** was calculated by first computing the circular mean of the population activity in each time window as

$$R = \frac{\sum_{j=1}^{N} r_j \exp\left(2\pi i \theta_j^*\right)}{\sum_{j=1}^{N} r_j}.$$

(9)

Here, $\theta_j^*$ is the preferred orientation (in radians) of neuron j. $r_j$ is the average activity of the neuron, which is calculated from the number of spikes emitted within a specific time window (10 ms in **Figures 6–8**). Only windows with a minimum number of active neurons (5 neurons emitting at least one spike for **Figures 6–8**) are included in the analysis. Population PO (pop. PO) is calculated as the angle of the resultant, $\arg(R)$, and the length of it, $|R|$, returns the population OSI (pop. OSI) (**Ringach et al., 2002**).

To quantify the transitions of population activity between different states, we developed a spontaneous transition (ST) index. This was calculated from histograms of population POs with high population OSI (OSI>0.5), $H(\theta)$. The histogram was obtained in the range from 0 to $\pi$, with bin sizes $\pi/24$ (as shown in **Figure 7A**). The OSI of $H(\theta)$ was, in turn, used to quantify the spread of specific states seen by the population activity:

$$Hosi = \left| \frac{\sum_{\theta} H(\theta) \exp(2\pi i\theta)}{\sum_{\theta} H(\theta)} \right|$$

(10)

The higher the $Hosi$, the more selective the distribution of highly selective population POs, meaning that population activity is concentrated around specific orientations. On the other hand, lower values of $Hosi$ imply that the distribution is less selective and hence wider specific states (preferred orientations) are visited by the population activity. We therefore used $(1 - Hosi)$ as a measure of switch between a wide range of specific states. We further multiplied that measure with the average population OSI (<pop. OSI>) to quantify how selective the visited states were on average, and defined the ST index as:

$$STI = (1 - Hosi) \times <pop. OSI>.$$

(11)

To evaluate how our results are affected by random transitions of the population activity and different levels of noise, we calculated a bootstrapped STI. For the bootstrap analysis, we shuffled the vector of activity of each neuron in time, and the ST index was recalculated for the shuffled population activity. This procedure was repeated for 100 times to obtain the mean and std of the shuffled estimate (plotted in red in **Figure 7B**). The bootstrapped measure of spontaneous transitions (bootstrapped STI; **Figure 8**) was calculated by subtracting the mean of the STI distribution obtained from the shuffled procedure, as the random baseline.

## Theoretical analysis

We start by analysing the dynamics of excitatory and inhibitory activity as described in **Equation 1**.

Solving for the stationary state solution of the responses we have:

$$\begin{aligned} r_E &= W_{EE}\, r_E \;+\; W_{EI}\, r_I \;+\; s_E \\ r_I &= W_{IE}\, r_E \;+\; W_{II}\, r_I \;+\; s_I \end{aligned}$$

(12)

The activity of inhibitory neurons can thus be expressed in terms of the inputs as:

$$
\begin{aligned}
r_E &= (I - W_{EE})^{-1}(W_{EI}\, r_I + s_E) \\
\Rightarrow r_I &= W_{IE}\,(I - W_{EE})^{-1}(W_{EI}\, r_I + s_E)\; +\; W_{II}\, r_I\; +\; s_I \\
\Rightarrow \left(I\; -\; W_{IE}(I - W_{EE})^{-1}W_{EI} - W_{II}\right) r_I &= W_{IE}(I - W_{EE})^{-1}s_E\; +\; s_I
\end{aligned}
\tag{13}
$$

where $I$ is the identity matrix. If we perturb the input to inhibitory neurons alone, the output change in the activity of inhibitory neurons can be calculated as:

$$
\delta r_I\; =\; \left[I - W_{EE}\,(I - W_{EE})^{-1}\,W_{EI}\; -\; W_{II}\right]^{-1}\delta s_I
\tag{14}
$$

Let us consider a simplified condition where excitatory and inhibitory populations have the same size, and the outgoing excitatory connections (to both E and I neurons; E→E and E→I) have a similar profile (which we refer to as W). Moreover, we assume that inhibitory outgoing connections have the same profile, up to an inhibition dominance factor of -g (that is, it can be expressed as -gW). *Equation 14* can now be expressed in terms of W and g as:

$$
\delta r_I = \left[I - W(I - W)^{-1}(-gW) + gW\right]^{-1}\delta s_I
\tag{15}
$$

Nonspecific/uniform patterns of perturbation to probe the presence of (nonspecific) inhibitory stabilized networks can now be described as a situation where the vector of inhibitory perturbation, $\delta s_I$, has more or less similar entries for all inhibitory neurons and, hence, perturbs the uniform mode of the network. Specific patterns of perturbation, in contrast, corresponds to the scenario where the pattern of perturbation is aligned with the specific eigenvector, $v^*$, of the network, corresponding to the eigenvalue, $\lambda$, of the weight matrix W:

$$
W\, v^*\; =\lambda\, v^*
\tag{16}
$$

For a one-dimensional ring network with specific connectivity according to a 1D feature of neuronal pairs (e.g. the difference of their preferred orientations), this specific mode entails a pattern of perturbation with stronger perturbation of neurons with similar preferred orientations. For networks of neurons with 2D receptive fields, where the weights of connections are modulated by RF similarity (assayed by RF correlations in our simulations), the specific mode can be described in a multi-feature space. To effectively perturb it in a 1D space, we can stimulate it by perturbing neurons in proportion to their RF correlation with a reference neuron.

Assuming such a specific mode exists in networks with feature-specific connectivity, *Equations 15,16* can be combined to write:

$$
\begin{aligned}
\delta r_I &= \left[I + W(I - W)^{-1}gW + gW\right]^{-1}v^* \\
\Rightarrow \delta r_I &= \left[1 + \lambda(1 - \lambda)^{-1}g\lambda + g\lambda\right]^{-1}v^* \\
\Rightarrow \delta r_I &= \tfrac{1-\lambda}{1-(1-g)\lambda}\, v^*.
\end{aligned}
\tag{17}
$$

$\lambda$ specifies the strength of the specific mode, arising from feature-specific connectivity, in the network. In feature-specific ISNs, the feature-specific connectivity between excitatory neurons, in the absence of specific inhibition to balance it, is unstable. This translates to $\lambda > 1$.

$g$ denotes the level of inhibition-dominance along the specific mode: $g = 1$ indicates the perfect balance; $g < 1$ leads to unstable dynamics along the specific mode; and $g > 1$ ensures a surplus of inhibition to keep the specific excitation in check by a dominant specific inhibition.

Networks with specific ISN properties can therefore be described, mathematically, by $\lambda > 1$ and $g > 1$.

We can therefore express $\lambda$ and $g$ by

$$
\begin{aligned}
g &= 1 + \alpha,\quad \alpha > 0, \\
\lambda &= 1 + \beta,\quad \beta > 0,
\end{aligned}
\tag{18}
$$

and rewrite the vector of changes in inhibitory activity in response to specific perturbations as:

$$\delta r_I = \frac{-\beta}{1 + \alpha\lambda} v^*. \tag{19}$$

$\frac{-\beta}{1+\alpha\lambda}$ analytically describes the slope of the line fitted to response changes versus input perturbations in our numerical simulations (e.g. as in *Figures 2*, *4*, *5* and *8*). Since $\frac{-\beta}{1+\alpha\lambda}<0$, this slope will be negative in specific ISNs. This means that perturbation of inhibitory neurons aligned with specific eigenvectors of the weight matrix changes the activity of inhibitory neurons in the *opposite direction*:

$$\delta r_I \;\propto\; -v^*. \tag{20}$$

## Acknowledgements

We thank NA Cayco-Gajic and DR Muir for comments on the manuscript. Code for reproducing main simulations and results are available from ModelDB (http://modeldb.yale.edu/259620).

## Additional information

### Funding

| Funder | Grant reference number | Author |
| --- | --- | --- |
| Biotechnology and Biological Sciences Research Council | BB/N013956/1 | Claudia Clopath |
| Biotechnology and Biological Sciences Research Council | BB/N019008/1 | Claudia Clopath |
| Wellcome | 200790/Z/16/Z | Claudia Clopath |
| Simons Foundation | 564408 | Claudia Clopath |
| Engineering and Physical Sciences Research Council | EP/R035806/1 | Claudia Clopath |

The funders had no role in study design, data collection and interpretation, or the decision to submit the work for publication.

### Author contributions

Sadra Sadeh, Conceptualization, Software, Methodology, Performing simulations, Writing the manuscript; Claudia Clopath, Supervision, Funding acquisition, Conceptualization, Writing the manuscript

### Author ORCIDs

Sadra Sadeh (iD) https://orcid.org/0000-0001-8159-5461
Claudia Clopath (iD) https://orcid.org/0000-0003-4507-8648

### Decision letter and Author response

Decision letter https://doi.org/10.7554/eLife.52757.sa1
Author response https://doi.org/10.7554/eLife.52757.sa2

## Additional files

### Supplementary files
• Transparent reporting form

### Data availability

All data generated or analysed during this study are included in the manuscript and supporting files.

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
