## [Decision Letter]

**Acceptance summary:**

New imaging and genetic techniques have created an opportunity to peer inside the brain's "black box". At the same time, current theory work in neuroscience has suggested that the brain operates in a particular dynamical regime that supports flexible and reliable computation. This manuscript suggests new experimental protocols that connect these tools with these theories to probe the hidden processing dynamics at play in the vertebrate brain. It is a thorough and detailed modeling study that will have a broad impact on how new experiments are planned and interpreted.

**Decision letter after peer review:**

Thank you for submitting your article "Patterned perturbation of inhibition is necessary to detect feature-specific inhibitory stabilization" for consideration by *eLife*. Your article has been reviewed by three peer reviewers, and the evaluation has been overseen by a Reviewing Editor and Michael Frank as the Senior Editor. The reviewers have opted to remain anonymous.

The reviewers have discussed the reviews with one another and the Reviewing Editor has drafted this decision to help you prepare a revised submission.

Summary:

This paper addresses how perturbation of neuronal activity can act as a probe to identify the dynamic properties of neuronal networks. The work focuses on "inhibition-stabilized networks (ISNs)" in which strong recurrent inhibition is required to stabilize strong recurrent excitation, and increased input to inhibitory neurons decrease both excitatory and inhibitory activities, called "the paradoxical effect." This manuscript investigates the response of inhibitory stabilized networks (ISNs) to "non-specific" and "patterned" perturbations of inhibitory cells in ISNs with feature-specific and non-specific connectivity rules. It aims to make explicit, experimentally testable predictions about such ISNs. The authors methodically demonstrate how such networks behave depending on the specificity of both the connections and perturbations to the network. Namely, they predict that inhibitory neurons will decrease their activity in response to patterned stimulation of inhibitory neurons when that perturbation aligns with neuronal receptive fields. They also show that ISNs with specific kinds of connectivity transition rapidly between different selectivity states. In general, the work is important and would potentially be of broad interest to both experimental and theoretically-minded readers.

Essential revisions, primarily computational:

While the theoretical investigation of the properties of ISNs that can guide the design of perturbation experiments is exciting and timely, the authors made several simplifying assumptions of network architectures and dynamics. Thus, the validity of the main results must be checked in more general and more biologically relevant settings:

1) Tuning widths, population sizes, connectivity: Request for more simulations:

As outlined above, this paper is attempting to make very clear connections to feasible experiments, which is fantastic to see. But, in general, we feel that the experiments reported in the manuscript build on top of each other before sufficiently establishing the generality and validity of the basic findings, and as a result, the same set of questionable model-assumptions are used throughout the paper. Specifically, equal numbers of excitatory and inhibitory cells with similar tuning properties are used in all simulations, but (1) It is known that there are fewer inhibitory neurons than excitatory neurons in real neural networks, (2) There are numerous papers that suggest, instead, broad tuning of inhibitory neurons. We would like to be convinced that the main conclusions will be supported by experiments using networks with biologically realistic architectures. Before exploring things like spiking networks, the authors should present results on more realistic inhibitory motifs with fewer inhibitory cells and broader inhibitory tuning.

The authors considered EI connectivity where all synaptic connectivity has the same tuning width. This seems too stringent as balanced states and ISN dynamics can be achieved with different spatial profiles of E and I connections. In that case, the relationship between the specific activity eigenmodes and connectivity or response similarity is less clear. Also, as pointed out in Figure 6—figure supplement 1B, the dynamic properties of ISNs are changed when E and I connectivities have different tuning widths.

2) Expanding to non-linear dynamics:

The main conclusion was derived mostly in the linear dynamics – the authors considered only a threshold-nonlinearity in the rate models and similarly in spiking networks as the f-I curve in I&F neurons is quite linear. Nonlinearities create interactions amongst the eigenmodes and may mitigate the "specific paradoxical effects." Also, in recently proposed networks with a supralinear transfer function, the dynamic range is divided into non-ISN and ISN regions by a "breakpoint" (Rubin et al., 2015). More simulations are needed to address how nonlinear dynamics might affect the conclusions of this work.

3) Exploration of stimuli that are the best perturbative probes of dynamics

The authors explore whether it is possible to design patterned perturbations based on stimulus response (a great thing to do!) and they report that some stimuli are not ideal for such experiments. Why not explore this further? If the goal is to make explicit experimental predictions, the authors should do a more detailed analysis of which stimuli would uncover the pattern of optical perturbation needed, and make that very clear to the reader.

Essential revisions, primarily textual:

1) Clarity of goals:

What are the actual goals of this paper? The way it is written, the central goal is to determine how optogenetic experiments could "reveal" specific ISNs (which is a slightly unclear goal in and of itself). But, then, what is the purpose of the final findings on transitions in selectivity states? On discussion does it become clear that the point of this paper is to more broadly determine what experiments could be done to test for the presence of ISNs with connection specificity in the brain. Both the perturbation results and the state transition results are effectively providing an experimental signature of such networks. But, this is not very clear as it stands. Moreover, once one realizes that the goal is to guide experiments (which is a great goal!), a host of other issues become apparent, as discussed in the following points.

2) Clarity for experimentalists:

Overall, we suspect that the paper would not be very easy to follow for an experimental neuroscientist. There is a great deal of jargon used throughout the paper, starting with the use of the term ISN in the fourth paragraph of the Introduction, without defining it for the reader. Other examples are references to eigenmode, selectivity state transitions, etc. This is all fine for computational readers, but given the goals of this paper, it would need a significant re-write to be interpretable for experimentalists. More broadly, the manuscript's clarity could be greatly enhanced, even for computational readers. For example, the authors write, "While uniform or non-specific perturbation was enough to reveal uniform ISNs, patterned perturbation as described above is necessary to reveal specific ISNs." What is meant by "reveal uniform/specific ISNs"? Do you mean "distinguish a specific ISN from a non-specific ISN"? Or something else? Throughout the paper, the language is confusing at times.

3) Additional Discussion sections:

Please discuss past ideas about 'detailed balance', i.e. Hennequin Vogels Gerstner 2012, Hennequin et al., 2014, Hennequin et al., 2017, and the inhibitory plasticity mechanism in Vogels et al., 2011.

Also, recent theoretical work suggested cortical circuits exhibit paradoxical effects because of disinhibitory loops rather than inhibition-stabilization mechanisms (Mahrach et al., BioRxiv 2019). This questions whether perturbations can reveal the dynamic properties in these complex, yet more biologically plausible networks. Please discuss the perturbation methods that can differentiate two prominent types of paradoxical effect mechanisms: disinhibitory loops vs. ISNs, and under which conditions similar conclusions as detailed in this paper can be derived.

---

## [Author Response]

Essential revisions, primarily computational:While the theoretical investigation of the properties of ISNs that can guide the design of perturbation experiments is exciting and timely, the authors made several simplifying assumptions of network architectures and dynamics. Thus, the validity of the main results must be checked in more general and more biologically relevant settings:1) Tuning widths, population sizes, connectivity: Request for more simulations:As outlined above, this paper is attempting to make very clear connections to feasible experiments, which is fantastic to see. But, in general, we feel that the experiments reported in the manuscript build on top of each other before sufficiently establishing the generality and validity of the basic findings, and as a result, the same set of questionable model-assumptions are used throughout the paper. Specifically, equal numbers of excitatory and inhibitory cells with similar tuning properties are used in all simulations, but (1) It is known that there are fewer inhibitory neurons than excitatory neurons in real neural networks, (2) There are numerous papers that suggest, instead, broad tuning of inhibitory neurons. We would like to be convinced that the main conclusions will be supported by experiments using networks with biologically realistic architectures. Before exploring things like spiking networks, the authors should present results on more realistic inhibitory motifs with fewer inhibitory cells and broader inhibitory tuning.

The reviewers are absolutely right that we need to show the specific paradoxical effect under more biologically realistic conditions, in terms of (1) the fraction of inhibitory/excitatory neurons and (2) broader tuning of inhibition.

To address the first issue, we simulated the model in Figure 2 with 20% inhibitory and 80% excitatory neurons, matching the fraction reported in the neocortex (Braitenberg and Schüz, 1998). To account for the decrease in the overall level of inhibition resulting from this decrease in the number, we increased the average weight of inhibition by a similar factor (4 times stronger), consistent with stronger weights reported for inhibition in the cortex (Hofer et al., 2011). Similar results were obtained for these networks, as shown in new Figure 2—figure supplement 4A.

We addressed the second issue by modifying the connectivity matrix between excitatory and inhibitory subpopulations. (Note that the broader tuning of inhibitory neurons does not matter in terms of the input, as we do not stimulate the network in response to external stimuli; the only inputs to inhibitory neurons are the patterned perturbations.) We allowed for a broader connectivity of I→E, I→I and E→I connections compared to E→E weights (by reducing m in the respective connections). We also decreased the weights such that the network still has spectral stability despite the decrease in specific inhibition. For stable networks with broader inhibitory connectivity, we obtained the same results, as shown in Figure 2—figure supplement 4B.

We combined the two conditions in another model, where the network had both fewer inhibitory neurons and broader inhibitory connectivity, and obtained similar results (Figure 2—figure supplement 4C).

We also tested whether our results hold in spiking networks with fewer inhibitory neurons and found that this is indeed the case (Figure 5—figure supplement 1). These results show that networks with fewer inhibitory neurons behave similarly in terms of their response to patterned perturbation of inhibition, as long as the weight of inhibition is strong enough to compensate for the smaller number of inhibitory neurons. Moreover, broad inhibitory connectivity is consistent with feature-specific ISNs, as long as the broadness does not compromise the stability of the network (i.e., the specific eigenmode of the spectrum does not become unstable due to lack of enough inhibition in that direction). We therefore concluded that our results also hold in networks with more biologically realistic properties of inhibitory neurons and their connectivity.

We added these results, figures and discussions to the revised manuscript.

The authors considered EI connectivity where all synaptic connectivity has the same tuning width. This seems too stringent as balanced states and ISN dynamics can be achieved with different spatial profiles of E and I connections. In that case, the relationship between the specific activity eigenmodes and connectivity or response similarity is less clear. Also, as pointed out in Figure 6—figure supplement 1B, the dynamic properties of ISNs are changed when E and I connectivities have different tuning widths.

To address this, we simulated networks with heterogeneous specificity of excitatory and inhibitory connections (Figure 2—figure supplement 5). We found that, although such a heterogeneous connectivity adds more noise to the observed pattern of response changes as a function of neuronal features, the specific paradoxical effect can still be statistically inferred from patterned perturbations (Figure 2—figure supplement 5C). Therefore, different profiles of E and I connections in terms of their functional specificity do not compromise the detection of specific ISNs.

2) Expanding to non-linear dynamics:The main conclusion was derived mostly in the linear dynamics – the authors considered only a threshold-nonlinearity in the rate models and similarly in spiking networks as the f-I curve in I&F neurons is quite linear. Nonlinearities create interactions amongst the eigenmodes and may mitigate the "specific paradoxical effects." Also, in recently proposed networks with a supralinear transfer function, the dynamic range is divided into non-ISN and ISN regions by a "breakpoint" (Rubin et al., 2015). More simulations are needed to address how nonlinear dynamics might affect the conclusions of this work.

We investigated the effect of response nonlinearity on our results by simulating rate-based networks with the following dynamics (same as in Ahmadian et al., 2013; Equation 2.3):

τdr/dt=-r+([Wr+s]_+)^n

where n determines the exponent of expansive nonlinearity (reported between 2 to 5 in experiments). Note that our previous results (obtained for threshold linear transfer functions) correspond to a special case with n = 1.

We assayed feature-specific paradoxical effects in networks with different values of n (1 to 4) and observed similar trends with all nonlinearities (Figure 2—figure supplement 6A). The expansive nonlinearity in fact seems to amplify the “specific paradoxical effect”, as inhibitory neurons with more negative perturbations show more positive response changes in networks with higher exponents. This is consistent with higher activation of excitation in networks with a higher exponent, and seems to reflect the fact that sharper nonlinearity amplifies the specific mode more in such networks (Figure 2—figure supplement 6B), which makes them more susceptible to nonlinearity (in fact, a network with n=5 showed signatures of such instability in terms of pathological oscillations; now shown). Thus, the main effect of nonlinear transfer functions seems to be exacerbating the specific eigenmode, and hence the specific paradoxical effect, as long as it does not compromise the stability. We also tested our results with smaller size of perturbations (up to 10 fold smaller) and found that feature-specific paradoxical effects could be observed in all nonlinear networks (not shown).

3) Exploration of stimuli that are the best perturbative probes of dynamicsThe authors explore whether it is possible to design patterned perturbations based on stimulus response (a great thing to do!) and they report that some stimuli are not ideal for such experiments. Why not explore this further? If the goal is to make explicit experimental predictions, the authors should do a more detailed analysis of which stimuli would uncover the pattern of optical perturbation needed, and make that very clear to the reader.

To address this highly relevant issue, we used naturalistic stimuli to probe the response correlations of neurons. We presented 200 natural images to the network and calculated the response correlation of neurons based on their responses to the ensemble of stimuli. We then repeated the patterned perturbations based on such response correlations of inhibitory neurons (with regard to different reference cells), and found that such naturalistic stimuli are indeed good candidates to reveal the feature-specific paradoxical effects (Figure 4—figure supplement 2). We added these results to the updated manuscript and discussed its consequences for experiments.

Essential revisions, primarily textual:1) Clarity of goals:What are the actual goals of this paper? The way it is written, the central goal is to determine how optogenetic experiments could "reveal" specific ISNs (which is a slightly unclear goal in and of itself). But, then, what is the purpose of the final findings on transitions in selectivity states? On discussion does it become clear that the point of this paper is to more broadly determine what experiments could be done to test for the presence of ISNs with connection specificity in the brain. Both the perturbation results and the state transition results are effectively providing an experimental signature of such networks. But, this is not very clear as it stands. Moreover, once one realizes that the goal is to guide experiments (which is a great goal!), a host of other issues become apparent, as discussed in the following points.

We rewrote the Introduction to clarify these points and relate our results better to experiments, also tried to make the language more transparent and accessible (see comments below).

2) Clarity for experimentalists:Overall, we suspect that the paper would not be very easy to follow for an experimental neuroscientist. There is a great deal of jargon used throughout the paper, starting with the use of the term ISN in the fourth paragraph of the Introduction, without defining it for the reader. Other examples are references to eigenmode, selectivity state transitions, etc. This is all fine for computational readers, but given the goals of this paper, it would need a significant re-write to be interpretable for experimentalists. More broadly, the manuscript's clarity could be greatly enhanced, even for computational readers. For example the authors write, "While uniform or non-specific perturbation was enough to reveal uniform ISNs, patterned perturbation as described above is necessary to reveal specific ISNs." What is meant by "reveal uniform/specific ISNs"? Do you mean "distinguish a specific ISN from a non-specific ISN"? Or something else? Throughout the paper, the language is confusing at times.

We hope that we have now addressed the issues raised above in the revised manuscript: we specifically corrected the points mentioned above, and, more generally, revised the manuscript to avoid jargon as much as possible, and to explain it when needed. We tried to relate our findings better and more clearly to the experiments. Moreover, we made the technical and computational language more precise and less confusing whenever needed.

3) Additional Discussion sections:Please discuss past ideas about 'detailed balance', i.e. Hennequin Vogels Gerstner 2012, Hennequin et al., 2014, Hennequin et al., 2017, and the inhibitory plasticity mechanism in Vogels et al., 2011.

That’s a great suggestion. We discussed that in the Discussion section.

Also, recent theoretical work suggested cortical circuits exhibit paradoxical effects because of disinhibitory loops rather than inhibition-stabilization mechanisms (Mahrach et al., BioRxiv 2019). This questions whether perturbations can reveal the dynamic properties in these complex, yet more biologically plausible networks. Please discuss the perturbation methods that can differentiate two prominent types of paradoxical effect mechanisms: disinhibitory loops vs. ISNs, and under which conditions similar conclusions as detailed in this paper can be derived.

We also added this to the Discussion section.